



# Crustal structures beneath the Eastern and Southern Alps from ambient noise tomography

Ehsan Qorbani[1,2], Dimitri Zigone[3], Mark R. Handy[4], Götz Bokelmann[2], and AlpArray-EASI working group[5]

[1]International Data Center, CTBTO, Vienna, Austria
[2]Department of Meteorology and Geophysics, University of Vienna, Austria
[3]Institut de Physique du Globe de Strasbourg, EOST, Université de Strasbourg/CNRS, Strasbourg, France
[4]Institute of Geological Sciences, Freie Universität Berlin, Berlin, Germany
[5]Eastern Alpine Seismic Investigation (EASI) AlpArray Complimentary Experiment. AlpArray Working Group

**Correspondence:** Ehsan Qorbani (ehsan.qorbani@univie.ac.at)

**Abstract.** We study the crustal structure under the Eastern and Southern Alps using ambient noise tomography. We use cross-correlations of ambient seismic noise between pairs of 71 permanent stations and 19 stations of the EASI profile to derive new high-resolution 3-D shear-velocity models for the crust. Continuous records from 2014 and 2015 are cross-correlated to estimate Green's functions of Rayleigh and Love waves propagating between the station pairs. Group velocities extracted
from the cross-correlations are inverted to obtain isotropic 3-D Rayleigh and Love-wave shear-wave velocity models. Our high resolution models image several velocity anomalies and contrasts and reveal details of the crustal structure. Velocity variations at short periods correlate very closely with the lithologies of tectonic units at the surface and projected to depth. Low-velocity zones, associated with the Po and Molasse sedimentary basins, are imaged well to the south and north of the Alps, respectively. We find large high-velocity zones associated with the crystalline basement that forms the core of the Tauern Window. Small-
scale velocity anomalies are also aligned with geological units such as the Ötztal and the Gurktal nappes of the Austroalpine nappes. Clear velocity contrasts in the Tauern Window along vertical cross-sections of the velocity model show the depth extent of the tectonic units and their bounding faults. A mid-crustal velocity contrast is interpreted as a manifestation of intracrustal decoupling in the Eastern Alps and decoupling between the Southern and Eastern Alps.

## 1   Introduction

Earth's crustal structure has been studied with classical regional earthquake tomography and active seismology for decades. However, gaining information on subsurface structure in seismically quiet areas has been challenging due to the lack of earthquake data and their infrequent occurrence and the high cost of active-source seismology. The emergence of seismic noise interferometry (e.g. Wapenaar, 2004; Shapiro and Campillo, 2004) has enabled seismologists to overcome this issue and to obtain more knowledge about Earth structures at various scales (e.g. Nicolson et al., 2012, and reference therein).

In this study, we investigate the crustal structure of the Eastern and Southern Alps (Fig.1) with ambient-noise tomography. The European Alps have resulted from N-S convergence of the Adriatic and European plates since the late Cretaceous and





Adria-Europe collision since late Eocene to Oligocene time (Schmid et al., 2004; Handy et al., 2010, 2015). The complex, non-cylindrical structure of the Eastern Alps and eastern Southern Alps (Schmid et al., 2013; Rosenberg et al., 2018) reflects the interplay between orogen-normal shortening and orogen-parallel motion during oblique indentation of Europe by the Adriatic microplate in Miocene time (Scharf et al., 2013; Handy et al., 2015; Favaro et al., 2017). Adria moved to the north and rotated counterclockwise with respect to Europe (Le Breton et al., 2017) such that indentation was partly accommodated by NNW-SSE shortening in the eastern Southern Alps (e.g. Schonborn, 1992) and partly by eastward tectonic escape of the Eastern Alpine orogenic crust (Ratschbacher et al., 1991a; Scharf et al., 2013; Schmid et al., 2013). This escaping orogenic crust is bounded by strike-slip faults (Periadriatic, Salzach–Ennstal–Mariazell–Puchberg (SEMP), Inn Valley, Mur-Mürz and Lavant Valley faults (e.g. Linzer et al., 2002; Schmid et al., 2004) and low-angle normal faults (Brenner, Katschberg faults) at either end of the Tauern Window (Selverstone, 1988; Genser and Neubauer, 1989; Scharf et al., 2013, 2016).

On the lithospheric scale several studies such as seismic tomography, anisotropy, and receiver functions have been done assessing the structures and proposing models of slab anomalies and geometry (Lippitsch et al., 2003; Schmid et al., 2004; Kissling et al., 2006; Mitterbauer et al., 2011; Karousová et al., 2013; Bianchi et al., 2014a; Handy et al., 2015; Qorbani et al., 2015; Hua et al., 2017; Rosenberg et al., 2018; Kästle et al., 2019). However regarding the crust, even though several seismological investigations mainly wide-angle reflection/refraction experiments (Bleibinhaus and Gebrande, 2006; Gebrande et al., 2006; Behm et al., 2007a; Brückl et al., 2007, 2010) have targeted the crust, the velocity structure of the Eastern Alps is still not fully understood. This was due to the low level of seismicity in the region and insufficient local earthquake data to perform traditional tomographic studies, as well as to the lack of active seismic experiments covering the region. Therefore, ambient-noise tomography appears to be perfectly suited to study crustal structure in this area.

Prior to this study, parts of the Alps and its surroundings were seismically imaged with noise-based tomography that provided Rayleigh-wave group-velocity maps (Stehly et al., 2009; Verbeke et al., 2012). These formed a database of surface-wave group and phase-velocity dispersion curves which were inverted to derive both group and phase-velocity maps of central Europe. Molinari et al. (2015) used the data base of Verbeke et al. (2012) to derive a 3-D shear-velocity model of the Alpine region and Italy. The Western Alps have also been studied with ambient-noise tomography (Fry et al., 2010) that yielded isotropic and anisotropic models of surface-wave phase velocity. Using surface-wave tomography from ambient noise and earthquake data, a shear-velocity model of the Alps (Kästle et al., 2018), and using ambient noise data shear-velocity model of the European crust and upper mantle (Lu et al., 2018) have been presented. To the east of the Eastern and Southern Alps (ESA), the crustal structure of the Carpathian-Pannonian region was studied with noise tomography depicted in surface-wave group velocity and 3-D shear-velocity maps (Ren et al., 2013). Behm et al. (2016) applied ambient-noise tomography to data from the ALPASS project to study the crust of the Eastern Alps, presenting Rayleigh and Love wave group-velocity maps and a shear-velocity model. However, the last two authors' results are limited to profiles used in those study.

Although the area of this study is included in the recent two shear velocity models (Kästle et al., 2018; Lu et al., 2018), little attempt has been made to interpret those velocity models with regard and in comparison to surface geology and smaller scale features. In this study, we therefore focus on the crustal structures of the Eastern Alps and Southern Alps. We present a local new high resolution 3-D shear-velocity model of the region using cross-correlation of seismic ambient noise. As an increment





to the recent shear-velocity models (e.g. Lu et al., 2018), we derive (group) velocity map for both Rayleigh and Love-wave and present separate Shear-velocity models out of Rayleigh and Love-wave velocities for the ESA. We then discuss our new models within the first 40 km of the crust with respect to the geologic and tectonic features.

## 2  Data and analysis

### 2.1  Ambient-noise Data

We used continuous 3-component seismic data recorded at 71 permanent broadband stations in the Eastern and Southern Alps – from the Seismic Network of Austria (OE, 1987), National Seismic Network of Switzerland (CH, 1983), Italian Seismic Network (INGV, 2006), Province Südtirol (SI, 2006), German Regional Seismic Network (GR, 2001), BayernNetz, Germany (BW, 2001), Slovenian Seismic Network (SL, 2001), Hungarian National Seismological Network (HU, 1992), and Slovak National Network of Seismic Station (SK, 2001). In order to improve ray coverage, we completed our dataset with 19 temporary broadband stations of the AlpArray-EASI project (AlpArray, 2015). The Eastern Alpine Seismic Investigation (EASI) is a collaborative seismological project as the first "AlpArray collaborative experiment" between the Swiss Federal Institute of Technology, University of Vienna (Austria), and the Academy of Sciences of the Czech Republic; it ran between July 2014 to July 2015. In that project, seismic stations were deployed along a North-South profile, roughly along longitude 13.5°E, from the internal Bohemian Massif to the Adriatic Sea with inter-station distances between 10-15 km. Using the 19 EASI stations improved station coverage especially in the Central Eastern Alps, where the station density is relatively low. After applying several selection criteria (see sec. 2.3.) on the computed cross-correlation functions (CC), 79 stations were selected for the tomography (see Fig. 1).

### 2.2  Waveform Pre-processing

The first step of any noise-based analysis requires pre-processing of the continuous waveform data, which strongly affects the quality of cross-correlation functions, dispersion curves, and the resulting velocity maps. Because noise characteristics and station configuration differ for each study, no universal pre-processing methodology exist. The best methodology and the various processing steps have to be tested for each dataset and are usually evaluated with basic parameters such as the symmetry of cross-correlation function, signal-to-noise ratio, and frequency bandwidth (e.g. Bensen et al., 2007; Poli et al., 2013). Here, we tested several classical pre-processing methods including windowing (e.g. Seats et al., 2012), whitening (e.g. Bensen et al., 2007) and one-bit normalization (e.g. Cupillard and Capdeville, 2010). The processing scheme chosen maximises the Signal-to-Noise ratio (SNR), defined here as the peak amplitude divided by standard deviation of noise. Our final pre-processing methodology follows Zigone et al. (2015) and consists of the following steps: 1) removing the instrument responses, high-pass filtering at 125 s, and glitch correction by clipping the data at 15 standard deviation. 2) Removal of the transient signals (e.g., earthquakes) by cutting the daily records into 2-hour segments on which we perform an energy test: when the energy of a segment is greater than twice the standard deviation of the energy of the 24-hour (daily) record, the 2h segment is removed.





3) Ambient noise is not spectrally white which may induce an amplitude bias in the resulting cross-correlations (e.g. Rhie and Romanowicz, 2004; Bensen et al., 2007). The noise spectrum is therefore normalized using a whitening function by dividing the amplitude by its absolute value between 1 and 100 s of period, without changing the phase. 4) We also perform a second clipping step in order to ensure that all the energy from transient sources that were not previously deleted by the energy test, such as small earthquakes, are properly removed from the waveforms. This is done by clipping amplitudes larger than four standard deviations of the whitened records. 5) Finally, the data are down-sampled to 4 Hz to reduce computational costs.

## 2.3 Computing cross-correlation functions

After pre-processing of the waveform, we compute cross-correlation functions (CC) for all stations pairs, which resulted in 4005 CC from 90 stations. The CC for each daily record and each station pair is computed over all possible combination of three-component data, vertical (Z), North-South(N), and East-West (E). This yields to nine inter-components, ZE, ZN, ZZ, EE, EN, EZ, NE, NN, NZ, constituting the correlation tensor. The ambient noise in the microseism frequency band is dominated by surface waves (Shapiro and Campillo, 2004; Shapiro et al., 2005); using all of these nine inter-components of CC enables us to construct both Rayleigh and Love waves from the computed CC.

Cross-correlations are computed in the frequency domain and then returned to the time domain to be stacked over the two years of data in order to reduce the seasonality of the noise sources. Rotation of the stacked CC was then performed based on the azimuth between the station pairs to obtain the CC on the radial (R), transverse (T), and vertical (Z) components, which form the nine inter-components correlation tensor, RR, RT, RZ, TR, TT, TZ, ZR, ZT, and ZZ. In order to select the clearer CC, we picked those that have signal-to-noise ratio (SNR) larger than 4. The SNR is calculated as max amplitude of the CC divided by the standard deviation of a noise window. We found that the CC with low SNR can be associated primarily with certain stations, marked in gray in Figure 1. We, therefore, removed those 11 stations, which resulted in 23% less CC. Figure 2 shows an example of two-year (2014-2015) stacked and rotated CC with the corresponding station-path shown in the top figure. Examples present short station-distances, between FETA-ROSI, and AAE31-ABTA, and also long distances, CEY-DAVOUX, and CONA-KNDS. Only ZZ, ZR, RZ, RR, and TT are shown in the figure as Rayleigh and Love waves are extracted from those 5 inter-components (Shapiro and Campillo, 2004).

The examples shown in Figure 2 were chosen based on the pair geometry and orientation with respect to the North Sea coastline, which should be the dominant noise source for our study region (Yang and Ritzwoller, 2008; Juretzek and Hadziioannou, 2016). We observe that on pairs parallel to the North Sea coastline, the CC are slightly asymmetric, with larger amplitudes for propagation directions arriving from north and northeast. This can be seen on CONA-KNDS (Fig. 2c) and on AAE31-ABTA (Fig. 2e), for which the surface wave amplitude is larger on the causal side. When station pairs are nearly normal to the North Sea coast line, such as CEY-DAVOX (Fig. 2b) and FETA-ROSI (Fig. 2d), we obtained strongly asymmetric CC. The differences between causal and acausal parts in the coast-normal station pairs is larger than the coast-parallel cases. Propagation directions from the North Sea coastline (from the northwest) results in large amplitude (from DAVOX to CEY, and from FETA to ROSI, Fig. 2). This supports the hypothesis that seismic ambient noise is mainly generated by the interaction of the ocean swells with the seafloor located in north and northwest of Europe. It also suggests that less energy is coming from southerly directions with





respect to the northerly directions. Figure 3 shows the nine inter-components correlation tensor (RR, RT, RZ, TR, TT, TZ, ZR, ZT, ZZ). The CCs are stacked in 2 km distance bins and sorted according to station distance. Rayleigh waves emerge from the RR, RZ, ZR, and ZZ, and Love waves from the TT. The cross terms (TR, RT, ZT, TZ) also carry some weak diffuse energy without any clear arrivals, which confirms the quality of the correlation component rotations. Note the slightly higher velocity for Love waves compare to Rayleigh waves.

## 3 Dispersion measurements

High-quality CCs are mandatory to measure accurate group-velocity dispersion curves. Quality checks of the CC based on SNR is one of the key tasks for preparing clear and reliable CC and to obtain acceptable empirical Green's functions (Bensen et al., 2007). That step removes poor-quality CC which is critical when using an automatic procedure for dispersion measurement. Figure 4 represents the signal-to-noise (SNR) of the folded acausal and causal CC with respect to inter-station distance (Fig. 4a) after selecting the CC with SNR larger than 4. Longer inter-station distances present lower SNR (Fig. 4a), which might be due to the lack coherent surface-wave energy in the noise wave field propagating between stations with long distance (Domingues et al., 2016). Variation of the SNR as a function of inter-station azimuth is shown in Figure 4b. Although the azimuthal distribution of the SNR could be frequency dependent, we found that the SNR is higher in the azimuth range between $90°$ to $120°$, and also between $270°$ to $320°$. These azimuths correspond to E-W to NW-SE inter-station orientation, which is nearly normal to the North Sea coastline. In order to improve the reliability of the dispersion measurements we stack the causal and acausal side of the correlation function to obtain a single CC for each station pairs (Bensen et al., 2007). Such procedure also broadens the frequency content of the merged CC by combining the different frequency content of opposite propagation directions (Yang and Ritzwoller, 2008; Shapiro and Campillo, 2004; Verbeke et al., 2012), which help the following travel time measurements.

We measured the group-velocity dispersion of the fundamental mode of the Rayleigh and Love waves for periods between 1 to 50 s using frequency-time analysis (FTA) (Levshin and Keĭlis-Borok, 1989). To improve the reliability of Rayleigh wave dispersions measurements we used the redundancy of the correlation tensor by using all components (RR, RZ, ZR, and ZZ) containing Rayleigh waves. FTA are first computed for each component $i$ independently, to obtain a normalized period-group velocity diagram $N_i(T, u)$, where $u$ is the group velocity and $T$ the period. Applying a logarithmic stacking in the period-group-velocity domain (Campillo et al., 1996), as $A_s(T, u) = \prod_i N_i(T, u)$, we then combined the four RR, RZ, ZR, ZZ components and formed the product of these four components for each station pairs, where the amplitude of $A_s(T, u)$ is dependent on the standard deviation of the group velocities. We evaluated the final dispersions on a $[A_s(T, u)]^{\frac{1}{i}}$ diagram, which allows us to obtain normalized period-group velocity diagram with amplitude between 0 and 1. The normalized period-group velocity diagram makes us able to select good quality dispersion measurements according to the amplitude. Here we selected the period-velocity values that have the maximum amplitude greater than 0.07. Examples of two period-group velocity diagrams are presented in Figure 5, for stations MOSI-SALO and ABTA-CONA. The dispersion curves are shown by the black line in the figure. The same procedure was applied to get the Love-wave dispersion curves using only the TT component.



To increase the quality of the velocity measurements, we applied a number of criteria: 1) To avoid high ray-path density in the central area with respect to other parts of the region, we removed all combination of temporary-temporary inter-stations and kept only the temporary-permanent pairs. 2) We removed all paths with the inter-station distances smaller than 2 wavelengths at each period. 3) We applied a second SNR pass (SNR > 5) for the correlations to pass the velocity measurements, to ensure that we obtain well-estimated travel times. 4) We exclude velocity measurements which were not in a range within two standard deviations from the mean velocity for a given period.

Applying those criteria, the dispersion data for each period were extracted, to be inverted to group-velocity maps. Figure 5 shows an example of velocity measurements obtained at 20 s of period for Rayleigh and Love waves. The stations and paths that satisfied our criteria are shown on the map. The measurements directly show high and low-velocity anomalies which are spatially stable, and related with the crystalline core zone of the Alps and the sedimentary basins, respectively.

## 4 Group-velocity tomography

We used the method of (Barmin et al., 2001) to invert the dispersion data and to derive tomographic images of surface-wave group velocity. This method is a damped least-square inversion based on minimization of a penalty function, which consists of a linear combination of data misfit, model smoothness, and magnitude of perturbations, including a Gaussian spatial smoothing function (see Barmin et al., 2001, for a detailed description). The magnitude of the model perturbation is controlled by two parameters, defined as $\lambda$ and $\beta$. If the ray coverage is relatively good, these two parameters do not affect the final model (e.g. Stehly et al., 2009; Poli et al., 2013; Zigone et al., 2015), which is the case for our study region. Therefore, we fix $\lambda$ and $\beta$ at 0.4 and 3, respectively. The spatial Gaussian smoothing is controlled by a damping factor and a width of smoothing area (also called correlation length in km). These parameters strongly affect the variance reduction of the final model. Stehly et al. (2009) recommended that the correlation length should be at least equal to grid size.

Using a grid size of 8 km, we tested several values for the correlation length ($\sigma$) and damping factor ($\alpha$), performing an L-curve analysis (e.g. Hansen and O'Leary, 1993; Stehly et al., 2009). Based on the variation of the variance reduction with respect to $\alpha$ and $\sigma$ for each period, and for Rayleigh and Love waves separately, we chose our optimized values between 28 and 40 for damping factor and between 20 to 26 for correlation length (Supplementary Table 1 shows the values selected for $\alpha$ and $\sigma$ for each period). The initial model of the inversion was the average group velocity at each period.

### 4.1 Resolution of tomography

We assessed the resolution of the tomographic inversion results:

1) Through the path density at each cell used for the inversion. Figure 6 presents the path density map at 8 and 16 s for both Rayleigh and Love waves. The path coverage of the region is good for most of the cells with a path density above 20. The path coverage reaches 100 paths per cell in the Southern Alps and in the Tauern Window, particularly for periods shorter than 15s. At the edges of the study region the resolution decreases rapidly due to fewer station and ray coverage (see Fig. 6).

2) By quantifying the dependence of the group velocity at each cell on the other cells (Barmin et al., 2001). That was done





via the resolution matrix, which depends mainly on the distribution of high-quality velocity measurement (path coverage, Fig. 6) and on the network geometry. The resolution is evaluated by plotting the resolution length defines here as the distance in kilometers, for which the value of the resolution matrix decreased to half. Figure 7 shows the map of the resolution length of

190 the final velocity model for Rayleigh and Love waves at 8 s and 16 s. At short periods (less than 15 s) most of the region shows resolution lengths between 8 and 15 km. This suggests that the smallest features that we are able to resolve are in the order of between 8 to 15 km, which corresponds to one to two cells in the final model. The resolution length increases for the longer periods due to the decreasing of the number of paths. Nevertheless, with respect to the study region, we are still able to map structures of about 50 km at the lower crust. Although the number of cross-correlations used to reconstruct the Love waves are

195 less than those for Rayleigh waves, the path density and resolution length obtained for the Love waves remain similar to those of the Rayleigh waves, which is sufficient to resolve the expected geological features.

## 4.2 Group-velocity maps

Figure 8 and 9 show the group-velocity maps at periods 5, 10, 15, and 20 s. In general, both Rayleigh (Fig. 8) and Love waves (Fig. 9) group-velocity maps present similar features, and correlate well with surface geology particularly at the upper crustal

depths. To assess the velocity pattern with respect to the geological units, we extracted their borders from the geological map of Austria (Egger et al., 1999) and the tectonic map of the Alps (Fig. 1, Schmid et al., 2004). The borders are shown as dashed lines in Figs. 8 and 9, representing the margins of the two Dolomite units to the South (Southern Limestone Alps, SLA) and to the north (Carbonates of the Northern Calcareous Alps, NCA), the crystalline core zone of the Alps (CZA) in between (Fig. 1), and the Tauern Window (TW) of the Eastern Alps (yellow dashed lines in Fig. 8 and 9). The CZA is well-marked by the

broad high-velocity anomalies extending from the west to the east of the region. The SLA and the NCA are also marked by low velocities. At 5 and 10 s period, the eastern border of the CZA matches with the group-velocity contrast (#1 in Fig. 8a). The southern margin of the CZA, particularly its western part is clearly fitted by the edge of the high-velocity anomaly in that area at 5 to 10 s (#2 in Fig. 8a).

A high-velocity zone is featured in the easternmost part of the CZA, at 5, 10, and 15 s. This feature might be associated with

210 the Koralpe, Wölz high pressure nappe system, the area on both sides of the Lavant Valley transform fault, which consists of eclogite facies and has the age of the Alpine tectono-metamorphic event of 90-110 Ma (Bousquet et al., 2008). The late Cretaceous (Eoalpine) Ötztal-Bundschuh and Silvretta metamorphic basement nappes are also perfectly imaged by high-velocity zones (shown as OTZ on Fig. 8a and b). At periods greater than 10-12 s, the OTZ no longer appears on the group-velocity map, which may indicate the depth extent of the OTZ. A high-velocity anomaly is observed in the western part of the TW, while its

eastern part shows lower velocities.

Figure 9 shows Love-wave group-velocity maps. As expected, Love wave presents higher velocities than Rayleigh wave, as noted by the difference between color scales of Figure 8 and 9. Similar to the pattern of the Rayleigh-wave group velocity, the CZA is well-marked by the Love wave high-velocity anomaly bounded by the two Dolomites provinces (Fig. 9a) to the north and to the south. The TW is also marked by high-velocity anomalies at most of the periods. However, similar to Rayleigh

waves, the western part of the TW shows higher velocity. There is a velocity increase to the west of the Po-Basin, at ∼11°E



(#3 in Fig. 9a) which becomes more pronounced at 10 s period. This might be associated with the depth extent of the magmatic rocks under the Southern Alpine sediments (Dolomites in the Trentino region, Italy). At 15 s period, a pronounced high-velocity anomaly occurs in the easternmost part of the NCA (#4 in Fig. 9c). A similar anomaly appears in the velocity model of (Behm et al., 2016). The Molasse and Po-Basin (Fig. 1) can be clearly located on both Rayleigh and Love group-velocity maps (Fig. 8 and 9). The OTZ metamorphic units are marked partly by the Love waves at 5 s period (Fig. 9a). Further discussion of this feature will be provided in Section 6 when describing the shear-wave velocity model.

## 5 Shear-wave velocity inversion

In order to derive a 3-D shear-velocity (Vs) model of the region, we performed Vs depth inversion using the linearized inversion procedure of Herrmann (2013). We first constructed local dispersion curves from the group-velocity maps at each cell (8x8 km) of the grid. These local dispersion curves are inverted to obtain local 1-D shear-velocity model at each cell which are finally combined to provide a 3-D shear-velocity model for the region. We excluded the group velocities of periods smaller than 4s since measuring group velocity of the fundamental mode of the surface waves at those periods could be easily mistaken with higher modes. As the inversion scheme is linearized, the accuracy of the final model strongly depends on the initial velocity model. To construct a good initial model, we used a three steps approach: (1) we extracted an average dispersion curve using all cells with more than 5 paths; (2) The average dispersion curve is inverted using a 1-D starting model proposed by Behm et al. (2007a); (3) Finally, the resulting average Vs model is used as starting model for the inversion of the local dispersion curves in each cell of the grid. The parametrization is made of 30 layers of 2 km thickness above a half space. Shear velocities range from 3 km/s in the top layer to 4.5 km/s in the half-space. The velocity is allowed to take a large range of values as long as the depth variation is smooth. We performed 30 iterations for the inversion, which was sufficient to achieve reasonable fit. The first two iterations were done with higher damping in order to not overshoot the model; the other 28 iterations were performed with a lower damping factor. As discussed below, the inversion results are well-defined solutions given the model parameterization. Figure 10 shows the average-velocity model obtained from Rayleigh and Love-wave average dispersion curves. We used this 1-D average-velocity models as the initial model to invert the local dispersion curves in order to obtain the best-fitting local 1-D velocity model at each cell. The distribution of misfit between the theoretical and estimated dispersion curves of the models at all periods are presented in Figure 10, showing small misfit usually below ±0.1 and ±0.25 km/s for Rayleigh (Fig. 10a) and Love waves (Fig. 10b) respectively. The higher misfit of the Love waves may come from the lower number of cross-correlations (only from TT inter-component). The depth resolution of the inversion can be assessed through the normalized resolution matrix of the computed model, which are shown in Figure 10 for the Rayleigh and Love average models. Both Rayleigh and Love waves allow a good resolution above ∼42 km depth, where the resolution matrices are symmetric (Fig. 10). The final shear-velocity models obtained from Rayleigh-wave group velocity are presented in Figure 11, and those from Love waves in Figure 12.



# 6 Results and Discussion

The shear-velocity maps (Fig. 11 and 12) show a number of features that match surface geology and tectonic units. In the following we discuss several interesting features by first focusing on the upper crust, and then on the lower crust. We will

approach each feature by first discussing the constraints from Rayleigh-waves Vs model (Fig. 11), and then the ones from Love-waves Vs model (Fig. 12). To clarify the discussion, a simplified geologic map (Egger et al., 1999) is shown in Supplementary Figure S1.

## 6.1 Upper crust; correlation with geology

Similarly to the Rayleigh-wave group-velocity maps, the Rayleigh-wave shear-velocity (hereafter RVs) model (Fig. 11) shows

a large high-velocity zone corresponding at the surface with basement units of the Tauern Window and Austroalpine units just north of the Periadriatic Fault. The high-velocity area is bounded by two lower-velocity zones at all depth slices which are associated with surface exposures of Mesozoic carbonates in the nappes of the Northern Calcareous Alps (NCA) and the Southern Alps (SA). The NCA corresponds to low shear velocities (< 3.1 km/s) down to 10 km depth (Fig. 12). To the south, the velocity model clearly separates the SLA from the CZA. At 4 to 10 km depth, the Dolomites of the Southern Alps show

velocities of < 3.1 km/s. The northern margin of the SA (dashed line in Fig. 11a, b, c) clearly matches the boundary between high and low velocities. The same pattern is observed on the Love-wave shear-velocity (hereafter LVs) model (Fig. 12).

A low-velocity anomaly (see the position as labelled "I" in Fig. 11a and 12a) is found under the Molasse Basin, with velocities for basinal sediments ranging from 2.6 to 2.9 km/s. At depths of > 4 km, this low-velocity zone extends beneath the NCA in accordance with the occurrence of a south-dipping Northern Alpine thrust fault that emplaced Mesozoic nappes of the NCA

onto Neogene Molasse sediments (e.g. Brückl et al., 2010). The anomaly appears to extend down to 10 km depth, indicating that the Neogene basinal sediments in the footwall of this fault form a wedge some 8-9 km thick (Steininger and Wessely, 2000; Hamilton et al., 2000). Alternatively, the anomaly extends no deeper than 8 km; the deeper part of this anomaly may be produced by downward smearing in connection with the prominent low velocity of the Molasse Basin. On the LVs model, the dominant low-velocity zones associated with the Molasse Basin (Fig. 12) extend down to about 8 km.

Another low-velocity anomaly in the northwestern part of the study area (labelled "II" in Fig. 11a and 12a) is more pronounced on the LVs model (Fig. 12). At shallow depths, it could be related to the Swiss Molasse Basin. However, the low-velocity zone extends down to 20 to 22 km depth, which may indicate a smearing effect with low velocity in the lower crust. Note that this anomaly is located at the edge of the study area where the ray coverage is relatively poor, particularly at longer periods. This leads to a decrease of the lateral resolution and the related Vs values.

A low-velocity anomaly in the Southern Alps labelled "III" in Fig. 11a and 12a with values of less than 3 km/s can be seen down to 10 km depth beneath the Po Basin of northern Italy. This basin contains several kilometers of Mio-Pliocene clastic sediments derived from the retro-wedge of the Alps and the pro-wedge of the northern Apennines (Merlini et al. 2002). The Po Basin is also easily identified on the LVs model (Fig. 12) in which the velocity values and depth of the low-velocity zone are more or less as in the RVs model.



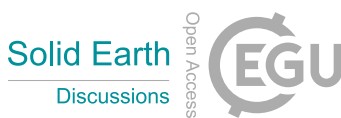

At the eastern part of the NCA, a small high-velocity anomaly (Anomaly "IV" , Fig. 11a and 12a) is observed at 4 and 6 km depth on the RVs model. It shows up along the southern margin of the NCA and might be associated with the eastern Greywacke zone, consisting primarily of Paleozoic low-grade metamorphic rocks. In the Greywacke zone the model exhibits velocities of more than 3.5 km/s. Such high velocities are no longer visible at 10 km depth. Similar high velocities have already been observed by (Behm et al., 2016) between the SEMP fault and the Northern Alpine thrust fault and also towards the

Bohemian Massif. They interpreted that feature as a southeastern tip of the Bohemian Massif dipping under the Alps. On the LVs model, this high-velocity zone is observed in a wider area. It can be traced down to 20 km depths as the anomaly becomes smaller.

Within the Austroalpine nappes, a clear velocity change (Anomaly "V", Fig. 11c and 12c) is seen on the RVs model to the east of the Tauern Window. Between 10 to 16 km depth, a high-velocity zone is found in the vicinity of the Lavant Valley fault,

which could be related to the Koralpe crystalline basement block. The area nearby between the Mölltal fault and the Koralpe unit is a relatively low-velocity zone (Fig. 11). This anomaly might be associated with the Gurktal Nappe, which comprises primarily low-grade Paleozoic rocks within the Austroalpine nappes.

One of the most notable features of the shear-wave velocity model (Anomaly "VI". Fig. 11b and 12b) is the higher velocity

in the western part of the Tauern Window compared to the eastern part. This is found at depths down to 10 km in the RVs model (Fig. 11a, b, c). The velocity contrast in the TW is more visible in the LVs model (Fig. 12a, b, c). The TW exposes both Penninic basement and underlying Subpenninic units (Schmid et al., 2013), with the latter containing high-grade basement (Central Gneiss) of the Venediger complex exposed in two domes at the W and E ends of the TW (Fig. S1 Egger et al., 1999; Schmid et al., 2013). The Western Tauern dome clearly corresponds to high velocities in our model. In contrast, the eastern

Tauern dome exhibits lower velocities. The Central Gneiss is subdivided geologically into subunits such as Granatspitz in the west and the Hochalm nappes in the east (e.g. Frisch et al., 1998; Schmid et al., 2013). However, it is not yet clear how such different lithologies could produce different shear-wave velocity in the TW region.

The shear-velocity models illustrate clearly the Silvretta and Ötztal-Bundschuh Nappes (OTZ) to the west of the Tauern Window (Anomaly "VII", Fig. 11a and 12a). These nappes are bounded by the Giudicarie Fault and the Engadin Window at its

eastern and western edges, and by the Inntal and Periadriatic Faults at its northern and southern margins, respectively (see Fig. 1 for fault locations). High shear velocity observed in this area might be associated with the Eoalpine high-grade metamorphic nappes, as seen on both Rayleigh and Love-wave shear-velocity models. The high velocity related to the OTZ is observed down to 6 km depth (Fig. 11b). Rocks of the Ötztal nappe underwent polymetamorphic metamorphism and deformation in Variscan and pre-Variscan time (e.g. Schuster et al., 2004). A notable feature here is the velocity contrast of the high and moderate

velocities on the Love-wave velocity model (at 4 km depth slice Fig. 12a, see also Fig 9a).

### 6.2 Lower crust

From depths 14-18 km downwards, we observe a clear separation between the high velocity under the TW and the low velocity beneath the OTZ. The high-velocity zone is cut by the Giudicarie Fault, which sinistrally offsets the Periadriatic Fault System





(Fig 11e). The Giudicarie Fault can be traced down to 40 km, confirming that it affects almost the entire crust (e.g. Pomella
et al., 2011, and references therein). However, the crust-mantle discontinuity (Moho) does not appear to be offset beneath
the Giudicarie Fault (Waldhauser et al., 2002; Spada et al., 2013), suggesting that this fault is a crustal feature that does not
penetrate down to the mantle lithosphere.

Towards the south, the Periadriatic Fault System (PAL) separates the Austroalpine nappes to the north from the Southern Alps,
including the Dolomites, to the south (Schmid et al., 2004). In the upper crust down to depth 14 km, most of the high-velocity
anomalies lie to the north of this fault. The PAL does not seem to separate units with different velocity structures. This may
indicate that units with similar physical properties are located on either side of the fault. On the Love-wave velocity model
(Fig. 12), we observe a clear, narrow high-velocity anomaly along the fault system. This feature is more pronounced along the
Giudicarie Fault (marked as "VIII" in Fig. 12g) perhaps due to magmatic rocks derived from crust and mantle along this fault
(von Blanckenburg and Davies, 1995; Rosenberg et al., 2004).

The low velocity associated with the Neogene sediments of the Po Basin is quite clear at 4 and 6 km depth. However, at 10 km
and deeper, the velocity reduction appears to deepen northward and extend to the lower crust (marked by "IX" in Fig. 12h).
Since this feature is not located exactly under the Po-Basin, it is unlikely that we have smearing effect due to the dominant
low velocity of the Po-Basin. Such a smearing effect can also be ruled out because it cannot be observed in such depth away
from the Po-Basin. Beneath the Po Basin, a high-velocity zone at > 22 km depth can be seen on both the RVs model and the
LVs model (marked "X" in Fig. 11h and 12h). The margins of this high-velocity domain are marked with red lines in Figure
13. They may show the boundary between intermediate and lower crust. More particularly, the southern margin might indicate
intermediate-lower crust boundary within the transition from thinned Dinaric crust to the Pannonian basin.

In Figure 13, we show depth slices at 30 and 40 km of our RVs and LVs models in comparison to the Vs model derived
from surface-wave phase velocity using a combination of ambient noise and earthquake data (Kästle et al., 2018), and also a
Rayleigh-wave Vs model derived from ambient noise data (Lu et al., 2018). The anomaly "X" is observed at 30 and 40 km is
marked by the red lines. It can be also observed on 30 and 40 km of the Kästle et al. (2018), Fig. 13e and 13g. To the south
of the PAL, the pattern of velocity changes of our model (Fig. 13a, b, c, d) and Kästle's model (Fig. 13e, g) are more-or-less
similar, however, they do not show similar pattern to the north of the PAL. Note that the Vs model of Kästle et al. (2018) has
been derived jointly from Rayleigh and Love waves, while that presented here has separate Rayleigh-wave Vs and Love-wave
Vs models. This could explain some discrepancies in pattern of the anomalies between our models and Kästle et al. (2018) to
the north of the PAL. The anomaly "X" can be seen in the Lu et al. (2018) on the 40 km depth slice (Fig. 13h). The relatively
low-velocity anomaly "IX", between the PAL and the anomaly "X" (Fig. 12h), can be also observed in the Lu's model (Fig.
13f, h). North of the PAL, our RVs model shows a low-velocity area at 30 and 40 km at the eastern part of the region. This can
be somewhat seen on the Lu et al. (2018) at 30 km. In addition, we see a clear high-velocity anomaly under the Tauern Window
at 40 km depth, while on the Lu et al. (2018) we do observe a broad low-velocity area at 40 km depth. It seems that to the north
of PAL where we have complex structures due to the interplay between orogen-normal shortening and orogen-parallel motion
(e.g. Ratschbacher et al., 1991a), our models resolve better small-scale velocity contrasts and features.





## 6.3 Cross-sectional view of the Vs model

Figure 14 shows cross-sections of the RVs model presented in Figure 11. Profile AA' crosses the Austroalpine (Silvretta and Ötztal-Bundschuh) nappe system and the Giudicarie Fault. The high-velocity associated with the Silvretta and Ötztal-Bundschuh basement units is observed down to 7-8 km depth, and then a layer of lower velocity anomaly can also be seen underneath this high-velocity anomaly. The high-velocity anomaly then shows a southeast-dipping continuation until it reaches a large high-velocity at > 20 km. Profile CC' (Fig. 14) crosses the easternmost part of the TW. This profile at longitude 13.3°E

is roughly parallel to the EASI profile (AlpArray-EASI, 2014). South of the Periadriatic Fault in the Southern Alps, a velocity change at about 20-25 km depth overlies that Moho depth at > 40 km (Behm et al., 2007a; Spada et al., 2013; Bianchi et al., 2015; Hetényi et al., 2018). To the north of the TW and under the Molasse basin, the velocity change is also observed, which may correspond to the boundary between the upper granitic and the lower mafic crust. Profile DD' (Fig. 14) crosses from the NCA across the Mur-Mürz fault to the Styrian basin. The high velocity in the vicinity of the Mur-Mürz fault in the uppermost

6 km seems to not be connected to any structure to the north. We also find a positive velocity change at about 10-15 km depth along the southern part of the profile. The crust-mantle boundary (Moho) beneath this area is shallow (Behm et al., 2007a), and becomes shallower to the east, toward the Pannonian basin (Horváth et al., 2006). Previously, a crustal thinning was proposed for this area (Horváth et al., 2006, and workers before). Behm et al. (2007b) also suggested a Moho jump of about 10 km under this area, which results in a Moho depth of 30 km. Such a sharp positive velocity change can be seen at 35 km depth on the

DD' (between 50 and 100 km horizontal distance on the profile), but is not clear further along the profile. Profile EE' (Fig. 14) crosses the length of the Tauern Window from E to W. The high-velocity zone beneath the TW is probably associated with the European basement and can be tracked at depth eastward along the profile. The lower velocity zone under the eastern part of the TW is also visible in the uppermost 15 km depth on this profile.

Figure 15 shows profile BB'. It is oriented N-S, coincident with the TRANSALP profile (TRANSALP Working Group, Ge-

brande et al., 2002), where sub-Tauern basement under the Tauern Window can be imaged as a high-velocity anomaly. A relatively low-velocity anomaly is observed beneath the Periadriatic Fault Zone at < 15 km depth. A geological interpretation of the TRANSALP profile (Schmid et al. (2004), see Bousquet et al. (2008)) is also shown in Figure 15. The pattern of the high-velocity zone associated with the sub-Tauern basement (marked with red dashed line) is in good agreement with the geological interpretation. A clear velocity contrast is observed at 20 km depth under the Inntal Fault and to the north. This could

be the European upper crust from the lower crust. To the south of Periadriatic Fault, there is a south-dipping velocity contrast reaching to 20 km depth; this may correspond with the boundary between upper granitic and lower mafic crust of the Southern Alps, somehow contrary to the geological interpretation shown in Figure 15.

## 6.4 Effect of anisotropy

The shear-velocity models extracted from Rayleigh and Love waves correlate well with most crustal geological and tectonic

units down to 20 km. However, we do observe some inconsistencies; for example, in the vicinity of the Mur-Mürz fault the LVs model shows higher shear velocity than the RVs model at most of the depths (Fig. 11, 12). Since we typically have very good





ray coverage for both Rayleigh and Love waves at lower periods (meaning shallower depths), and therefore greater resolution, the difference between the Rayleigh and Love waves, especially in the uppermost 20 km of crust, is noteworthy. As Love and Rayleigh waves are sensitive to shear displacement in different orientations (horizontal versus vertical), different velocity
anomalies between Rayleigh and Love, and particularly high-velocities of the Love waves may indicate seismic anisotropy. The observed velocity difference may be attributed to preferred alignment of the main schistosity and shear zones subparallel to the shearing plane of the Mur-Mürz Fault.

The velocity difference between the western and the eastern parts of the Tauern Window (Anomaly "VI" in Figs. 12) might be related to the difference in the orientations of the anisotropy. Structural data from deeply exhumed Penninic and Subpenninic
units indicate that the main schistosity strikes NE-SW and is subvertical in upright, post-nappe folds of the west, whereas it is variably oriented to subhorizontal in folds in the east TW (Scharf et al., 2013; Rosenberg et al., 2018).

We also found a striking velocity difference in the RVs and LVs models to the east of the Tauern Window at > 20 km depth (Fig. 12). This occurs in the eastwardly extruded block of orogenic crust (Alcapa) bounded by the aforementioned sinistral SEMP and dextral Periadriatic Faults. Eastward, orogen-parallel escape of the Alcapa in Miocene time is attributed to a combination
of indentation of the Adriatic plate and pull in the upper plate of the retreating Carpathian orogen (e.g. Royden and Baldi, 1988; Ratschbacher et al., 1991a; Horváth et al., 2006; Favaro et al., 2017, and references therein). The velocity contrast at the western margin of the high-velocity domain (yellow line in Fig. 13a) is quite discordant to the trend of Moho depth contours (Spada et al., 2013) and might represent the boundary between thinned intermediate crust and lower crust of Alcapa. We interpret this velocity contrast as a possible zone of intracrustal decoupling at the base of the laterally eastward extruded
Alcapa unit. The observed velocity difference between Love-wave Vs model and Rayleigh wave model may reflect shear-induced anisotropy originating from eastward motion of the Alcapa block above the subducting lithosphere during Miocene Adria-Europe convergence. Such high velocity is not seen in the (Kästle et al., 2018) Vs model, possibly due to the fact that their model is isotropic and jointly inverted from Rayleigh and Love dispersions. This inversion may have averaged any anisotropic effects due to contrasting Rayleigh-Love Vs differences. We do observe a slightly low-velocity anomaly to the
east of the TW on the Rayleigh-wave model of Lu et al. (2018). Since they only presented Rayleigh-wave Vs model, it is not possible to discuss and compare the anisotropy effect on the RVs and LVs difference from the Lu et al. (2018).

## 7 Conclusions

We used two years of ambient noise data recorded at a set of permanent and temporary stations in the Eastern and Southern Alps with an average station spacing of 232 km in order to perform ambient noise tomography and to derive a local high
resolution Vs model of the crust. As an increment to the previously presented Vs model for the Alps (Kästle et al., 2018), and for Europe (Lu et al., 2018), we presented here both Rayleigh-wave and Love-wave shear-velocity models. Our high-resolution 3-D shear-velocity models show very good correlation between the velocity contrasts and geology projected to depth from the surface. The models reveal details of the crustal structure down to depth of 40 km in higher resolutions that the previous Vs models (Kästle et al., 2018; Lu et al., 2018) have not shown and discussed.

The observed high-velocity anomalies are associated mainly with the crystalline core zone of the Alps, whereas the sediments of the Northern Calcareous Alps and the Southern Alps generally coincide with low velocities. The Molasse and Po Basins also correlate with low-velocity anomalies. Individual tectonic units (e.g. Silvretta and Ötztal-Bundschuh nappes, Koralpe unit) are also delimited by velocity contrasts. A velocity contrast at 20-25 km depth found mainly south and north of the TW (profiles BB' and CC', Figs. 13, 14) perhaps represents a boundary between the upper and the lower crust. The high velocity

and velocity contrast observed at depth > 20 km to the east of the TW can be interpreted as an intracrustal decoupling horizon that accommodated east-directed, orogen-parallel lateral extrusion of orogenic crust above lithospheric subduction during N-vergent Adria-Europe convergence.

Presenting separate Rayleigh and Love-wave Vs models made us able to observed a number of discrepancies between the Rayleigh and Love-wave shear velocities, e.g., around the Mur-Mürz fault. That may be attributed to strain-induced orientation

of the dominant foliation subparallel to the fault planes. Future studies of anisotropy are required to constrain the depth extent of this anisotropy, for example, by jointly inverting the Rayleigh- and Love-wave dispersions to construct an anisotropic shear-velocity model of the region.

*Acknowledgements.* We thank R. Schuster, E. Kissling, I. Bianchi, and E. Brückl for helpful discussions and all colleagues from IG Prague, University of Vienna, and ETH Zürich involved in the EASI seismic profile. A complete list of people who contributed to the EASI project

is provided at http://www.alparray.ethz.ch/. We thank B. Bahrami for digitizing the features and margins of the geological units and velocity anomalies superimposed on the tomographic images were digitized by QGIS. This work was partly supported by the Austrian Science Foundation (FWF), projects 26391 and 24218. The authors also thank the Austrian Agency for International Cooperation in Education & Research (OeAD-GmbH) for funding the Amadée project, FR02/2017. This project was co-funded by the French Ministry for European & Foreign Affairs and the French Ministry of Higher Education and Research, project number PHC-AMADEUS 38147QH. Thanks go to

the IPGS for its support of D. Zigone via the 2016 IPGS-internal call, and to the German Science Foundation (DFG) for its support of M.R. Handy (projects Ha 2403/19, 20). Finally, we acknowledge the Seismological Networks of Austria (ZAMG), Switzerland (CH), Italy (INGV), Südtirol (SI), German Regional Seismic Network (BGR), Bavaria (BayernNetz), Germany (BW), Slovenia (ODC), Hungary (HU), and Slovakia (SK) use of data as made available through the GFZ webdc data center, http://eida.gfz-potsdam.de/webdc3/.

*Disclaimer.* The view expressed herein are those of the authors and do not necessarily reflect the views of the CTBTO Preparatory Commis-

sion.

*Data availability.* Data from Seismological Networks of Austria (ZAMG), Switzerland (CH), Italy (INGV), Südtirol (SI), German Regional Seismic Network (BGR), Bavaria (BayernNetz), Germany (BW), Slovenia (ODC), Hungary (HU), and Slovakia (SK) are available through the GFZ webdc data center, http://eida.gfz-potsdam.de/webdc3/.





AlpArray-EASI Team: Jaroslava Plomerová, Helena Munzarová, Ludek Vecsey, Petr Jedlicka, Josef Kotek, Götz Bokel-
mann, Irene Bianchi, Maria-Theresia Apoloner, Florian Fuchs, Patrick Ott, Ehsan Qorbani, Katalin Gribovszki, Peter Kolinsky,
Peter Jordakiev, Hans Huber, Stefano Solarino, Aladino Govoni, Simone Salimbeni, Lucia Margheriti, Adriano Cavaliere, Edi
Kissling, John Clinton, Roman Racine, Sacha Barman, Robert Tanner, Pascal Graf, Laura Ermert, Anne Obermann, Stefan
Hiemer, Meysam Rezaeifar, Edith Korger, Ludwig Auer, Korbinian Sager, György Hetényi, Irene Molinari, Marcus Herrmann,
Saulé Zukauskaité, Paula Koelemeijer, Sascha Winterberg.

*Author contributions.* EQ performed data preparation, analysis, and velocity inversions. EQ also prepared the manuscript. DZ provided most
of the codes used in the analysis. MH and GB were participated in the geological interpretations. All authors also contributed to reviewing
and editing the manuscript. GB provided the financial support of the work.

*Competing interests.* The authors declare that they have no conflict of interest.





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



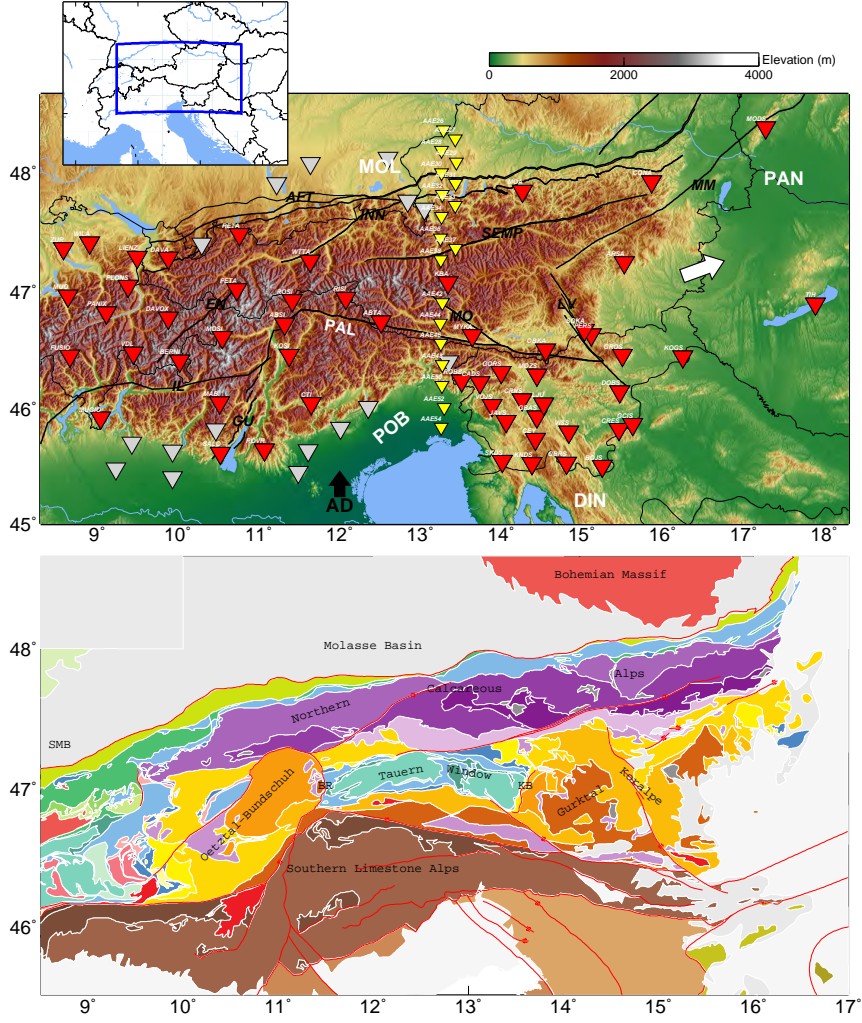

**Figure 1.** Study region in the Eastern and Southern Alps. Top: Permanent stations are marked in red and grey, and the 19 EASI stations used are shown in yellow. Main faults of the region are represented by black lines. The faults are indicated as AFT: Northern Alpine Front Thrust, INN: Inntal, SEMP: Salzach-Emdtal-Mariazell-Puch, MM: Mur-Mürz, LV: Lavant Valley, MO: Mölltal, GU: Giudicarie, IL: Insubric, EN: Engadine, PAL: Periadriatic line. MOL: Molasse basin, PO: Po-basin; PAN: Pannonian basin. DIN: Dinarides mountain belt. Black arrow shows the convergence vector of the Adriatic Plate (AD) with respect to the European Plate (EU). A white arrow illustrates the direction of eastward escape of the Alcapa tectonic block (see section 6.4). After the different selection criteria, the stations shown in red and yellow entered into the inversion (see text). Bottom: Tectonic map of the study region (Schmid et al., 2004, 2008, from http://www.spp-mountainbuilding.de). Units discussed in the text are labeled on the map. SMB: Swiss Molasse Basin, BR: Brenner fault, KB: Katschberg fault. Red lines represent the faults.



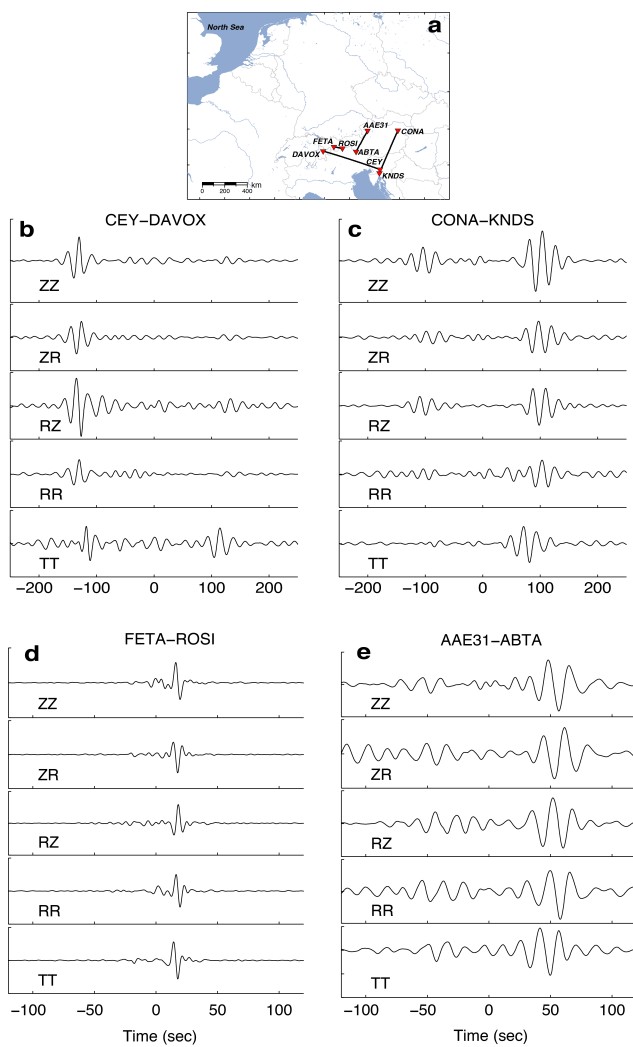

**Figure 2.** Examples of stacked and rotated cross-correlations (CC) from two years of data (2014-2015). The top figure shows the inter-station paths and geometry with respect to the North Sea coastline. FETA-ROSI, and AAE31-ABTA represent short station distance while CEY-DAVOUX, and CONA-KNDS are in the long distance.



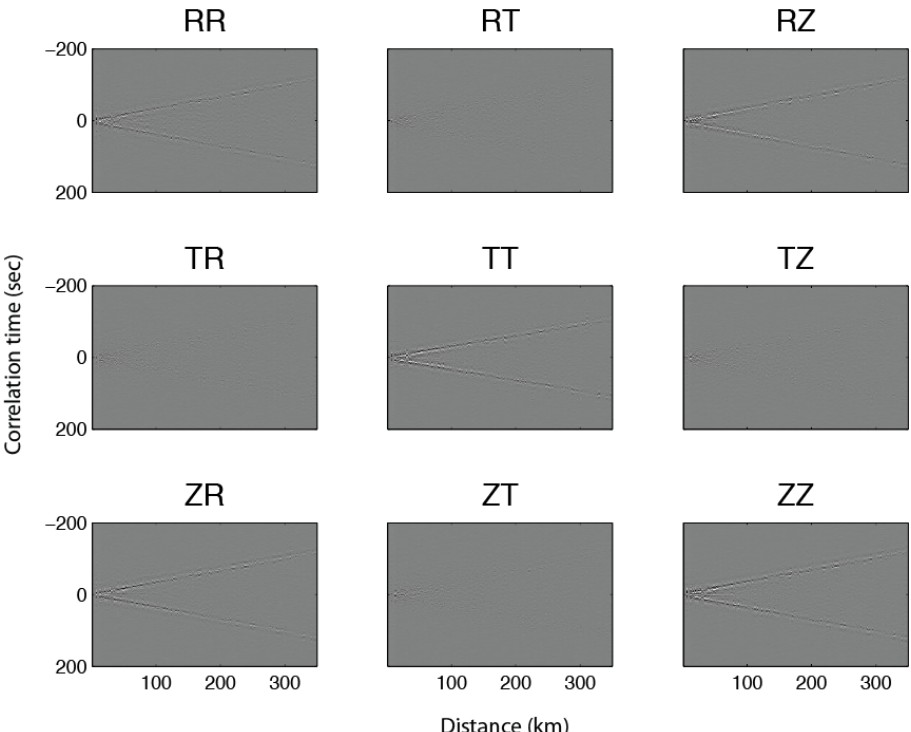

**Figure 3.** Correlation tensor including nine inter-components (RR, RT, RZ, TR, TT, TZ, ZR, ZT, ZZ). The tensor shows the surface waves between all station pairs. Rayleigh waves appear on the RR, RZ, ZR, ZZ, and Love waves on the TT. Note the slightly faster arrival of the Love waves with respect to the Rayleigh-wave.

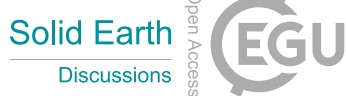

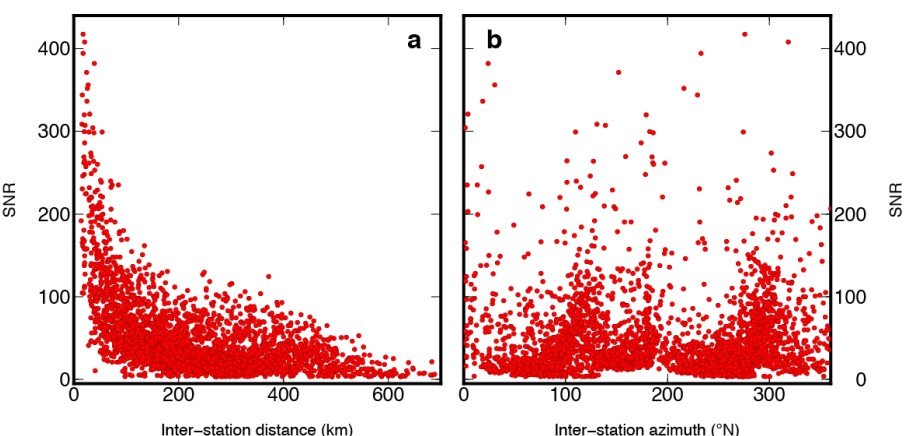

**Figure 4.** Signal-to-noise ratio (SNR) of the cross-correlations. a) The SNR of the stacked acausal and causal CC with respect to inter-station distance. b) The SNR as a function of inter-station azimuth. Lower SNR is observed for longer inter-station distances (also see the text).

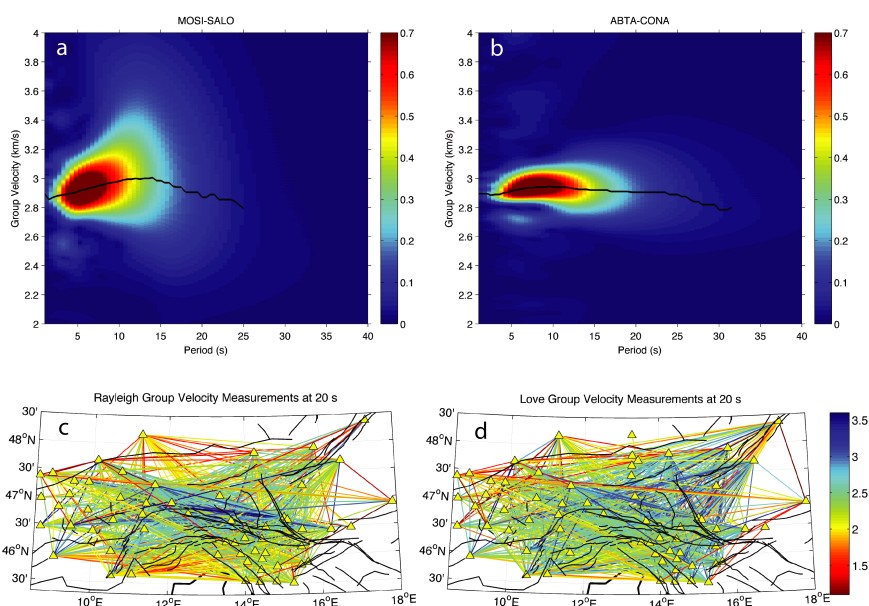

**Figure 5.** Top: Example of normalized period-group velocity diagram from a) MOSI-SALO and b) ABTA-CONA. The black line represents the extracted dispersion curve. Bottom: Example of velocity measurements at 20 sec period for both c) Rayleigh waves and d) Love waves. Note that stable high and low-velocity zones can be seen already in the velocity measurements.



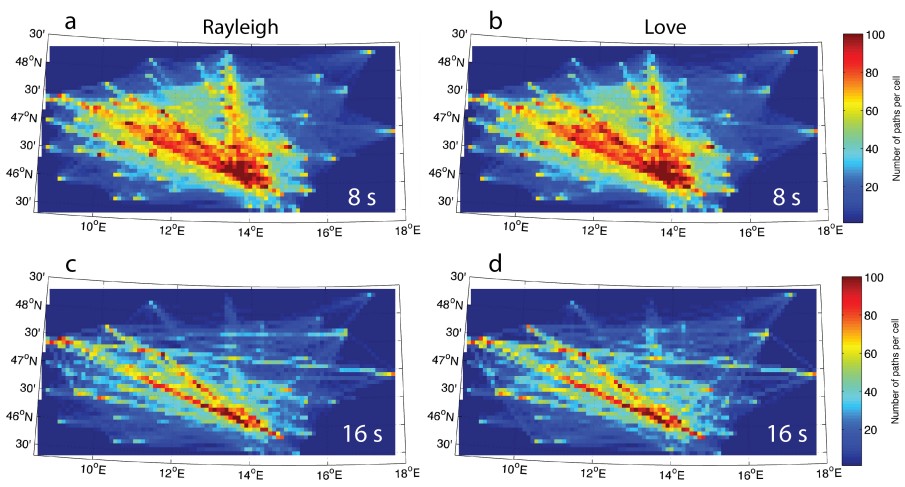

**Figure 6.** Path density map for the tomography inversion: a, c) for Rayleigh waves at 8 and 16 s respectively, and b, c) for Love waves at 8 and 16 s. The path coverage is generally good for the entire region Most of the cells have path density more than 20 rays per cell.





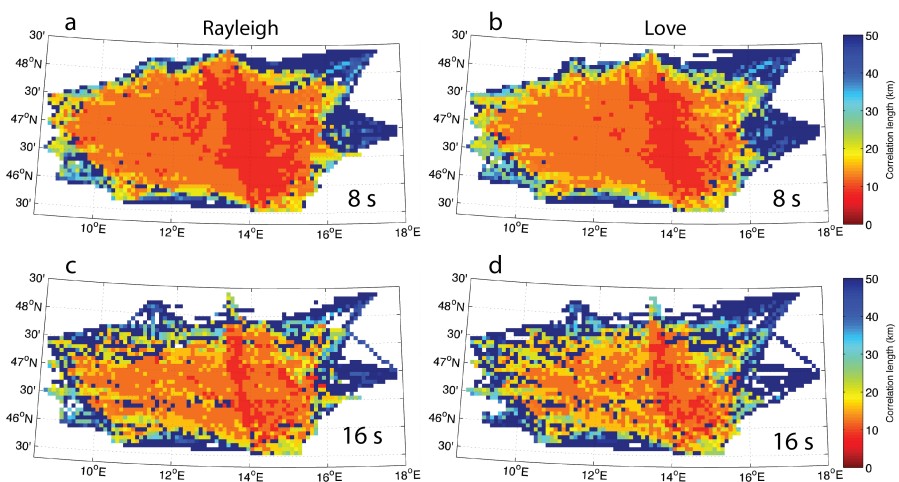

**Figure 7.** Resolution of the tomographic inversion shown via resolution length maps, for the final model. Colors show the resolution length, e.g. the distance for which the value of the resolution matrix decreases to half. a, c) for Rayleigh waves and b, c) for the Love waves at 8 and 16 sec. The resolution length indicates a spatial resolution of 8 and 50 km, with most of the region around 15-20 km.



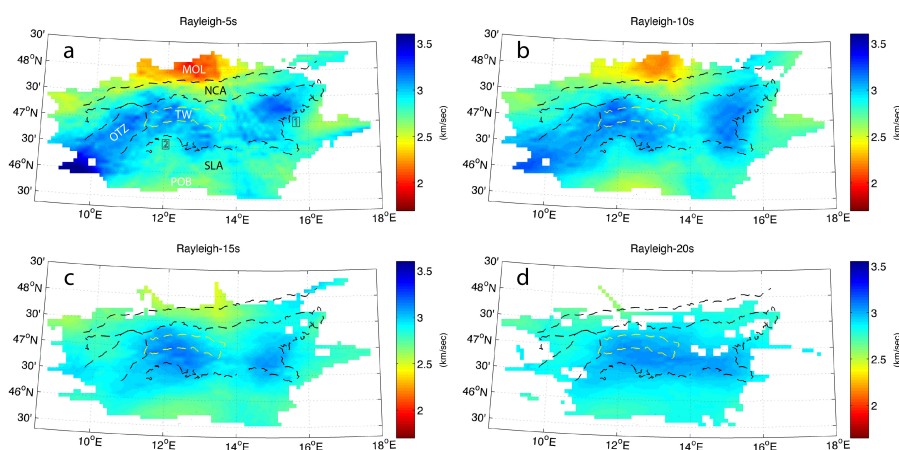

**Figure 8.** The obtained Rayleigh-wave group-velocity maps at periods 5, 10, 15, and 20 s. Dashed lines (Egger et al., 1999; Schmid et al., 2004) represent the margins of the Northern Calcareous Alps (NCA), the crystalline core zone of the Alps (CZA), the Ötztal block (OTZ), the Tauern Window (TW). MOL: Molasse Basin, and POB: Po-Basin. The western margin of the CZA (#1 at 5 s) and the northern margin of the SLA (#2 at 5 s) are well-marked by the velocity contrast at 5 s.

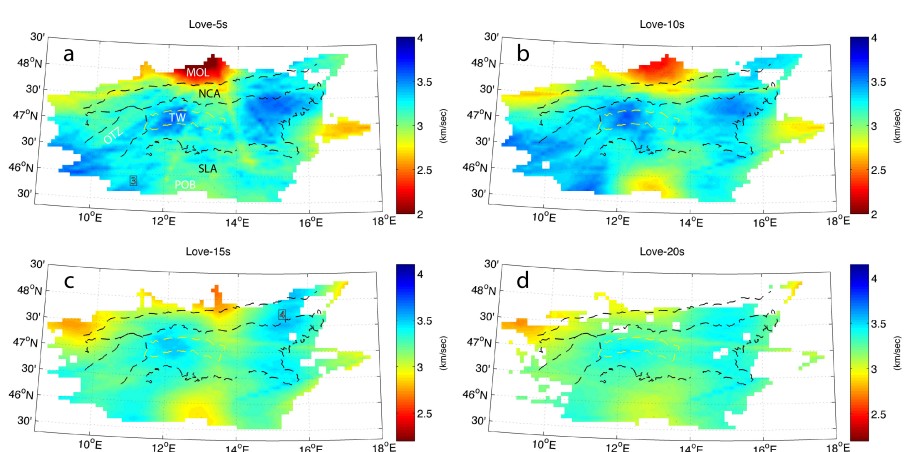

**Figure 9.** The obtained Love-wave group-velocity maps at periods 5, 10, 15, and 20 s. See Fig. 9 caption for abbreviations. #3 at 5 sec shows the high-velocity anomaly to the west of the Po-Basin. #4 at 15 sec shows a notable high-velocity anomaly in the easternmost part of the NCA.




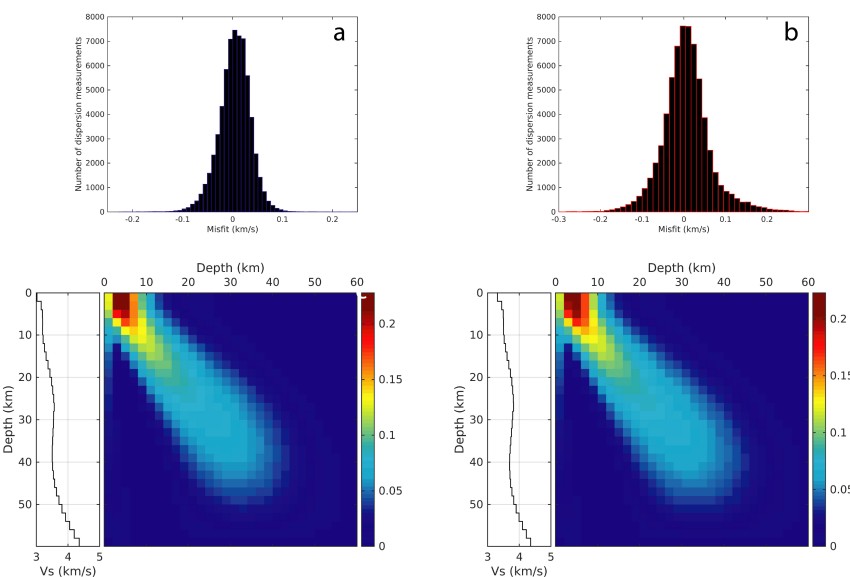

**Figure 10.** Histograms of the distribution of misfit between synthetic and observed dispersion curves for Rayleigh waves (a) and Love waves (b) shear-velocity model at all periods. The misfit of the Rayleigh-wave model (RVs) is generally less than 0.1 km/s, and for the Love-wave model (LVs) less than 0.25 km/s (see text). c) the resolution matrices of the average-velocity model derived from Rayleigh waves and d) from the Love wave. The left panel on each show the average 1-D-velocity model for the region.



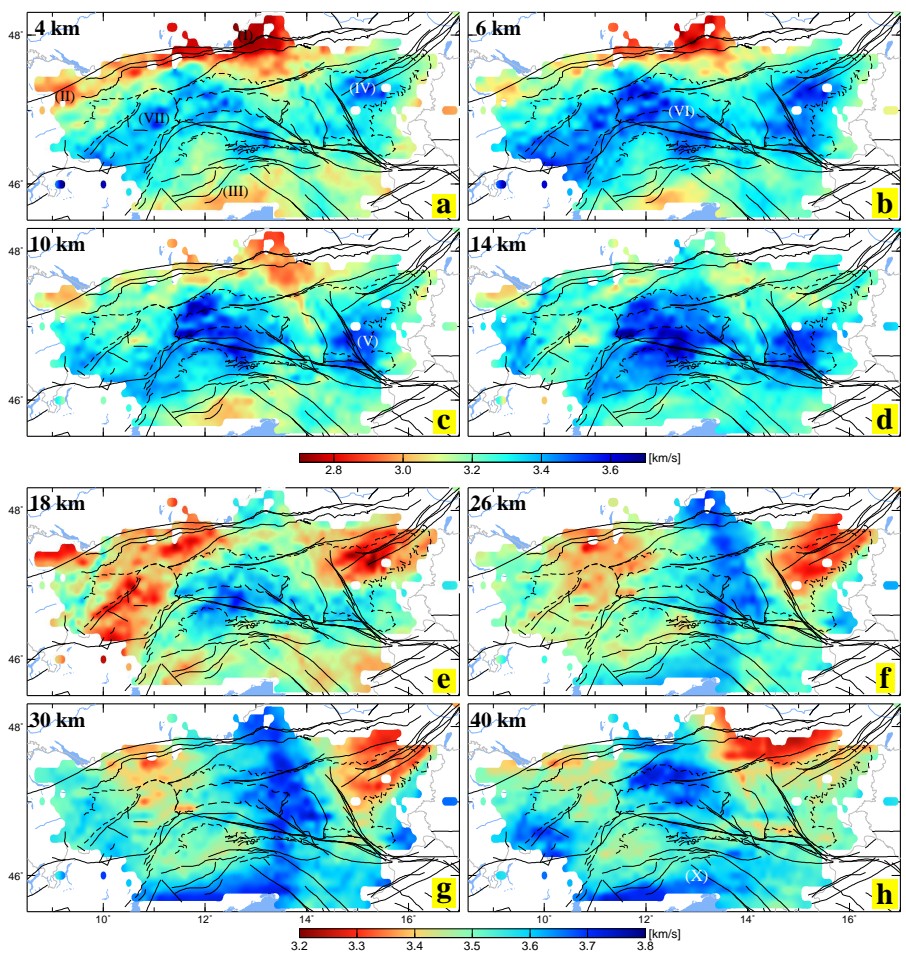

**Figure 11.** Shear-velocity model derived from inversion of the Rayleigh-wave group-velocity maps. Black lines represent the main faults in the region (modified from Schmid et al. (2004)). Dashed lines show the main geological units of the region (Geological map of the Eastern Alps, Egger et al. (1999)) to be compared with the velocity patterns, together with number indicating the geographical regions discussed in the discussion section. See text for the velocity anomalies marked by Roman numerals.



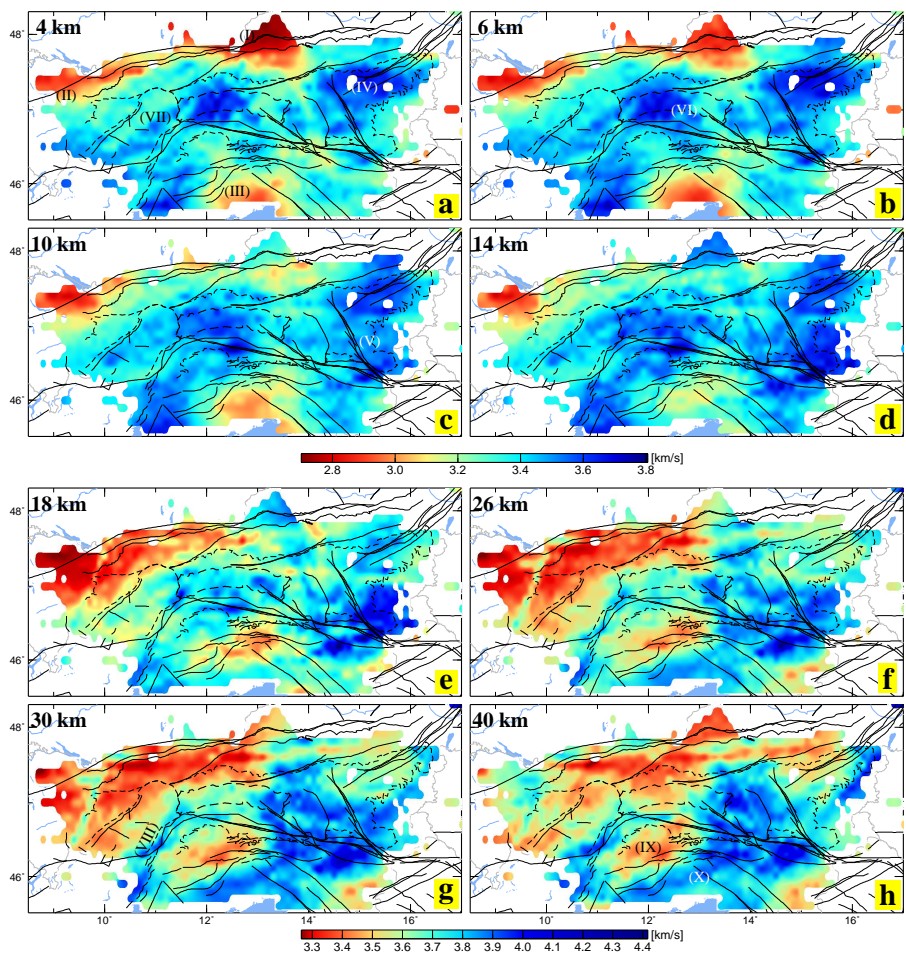

**Figure 12.** Shear-velocity model derived from inversion of the Love-wave group-velocity maps. Solid and dashed black lines as on previous figure. See text for the velocity anomalies marked by Roman numerals.



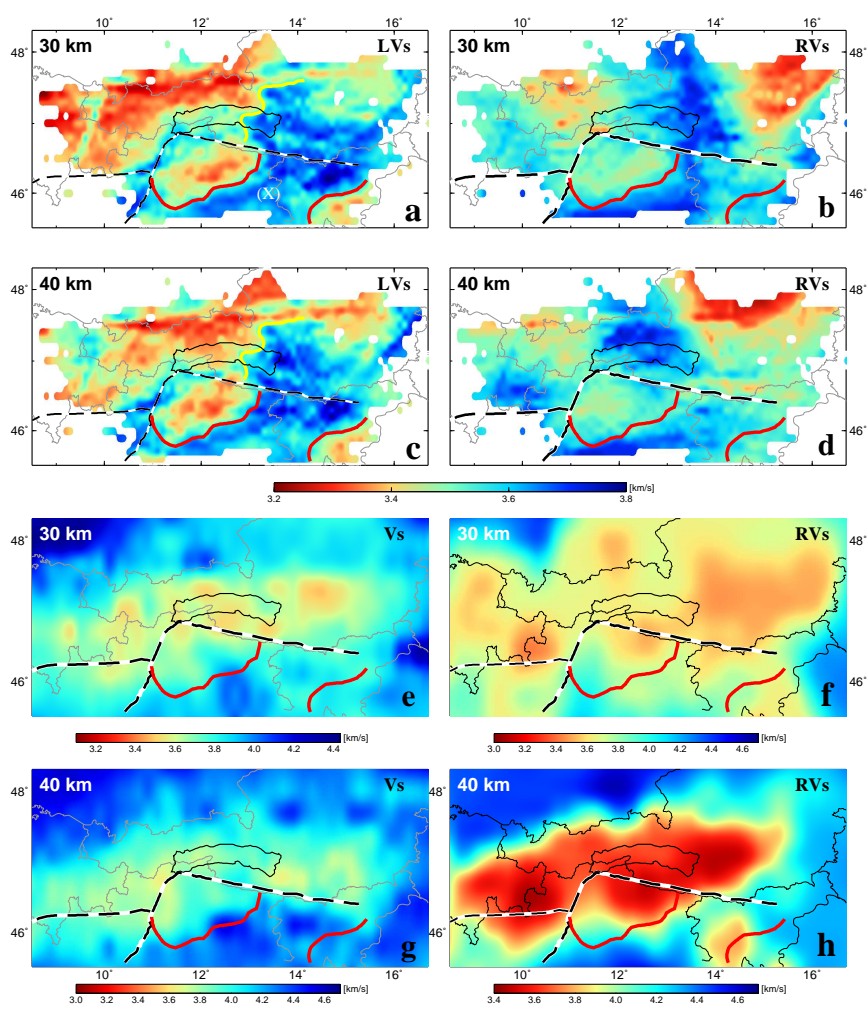

**Figure 13.** a & c: Depth slices of the Love-wave shear-velocity model (LVs), and b & d: Rayleigh-wave Vs model (RVs) presented in this study. The red lines outline the high-velocity anomaly to the south of the Periadriatic line (PAL) marked by "X". Periadriatic and Giudicarie faults are shown black-white lines labelled PL and GF, respectively. The western margin of the high-velocity to the north of the PAL is marked by the yellow line. e & g: Depth slices of the Vs model of Kästle et al. (2018) at 30 and 40 km depth respectively. f & h: Depth slices of the RVs model of Lu et al. (2018) at 30 and 40 km depth. The high-velocity anomaly marked by "X" on our Vs models can also be seen on the Kästle's Vs model (e and g) and partly on the RVs of Lu's model (h).

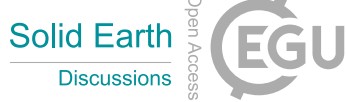

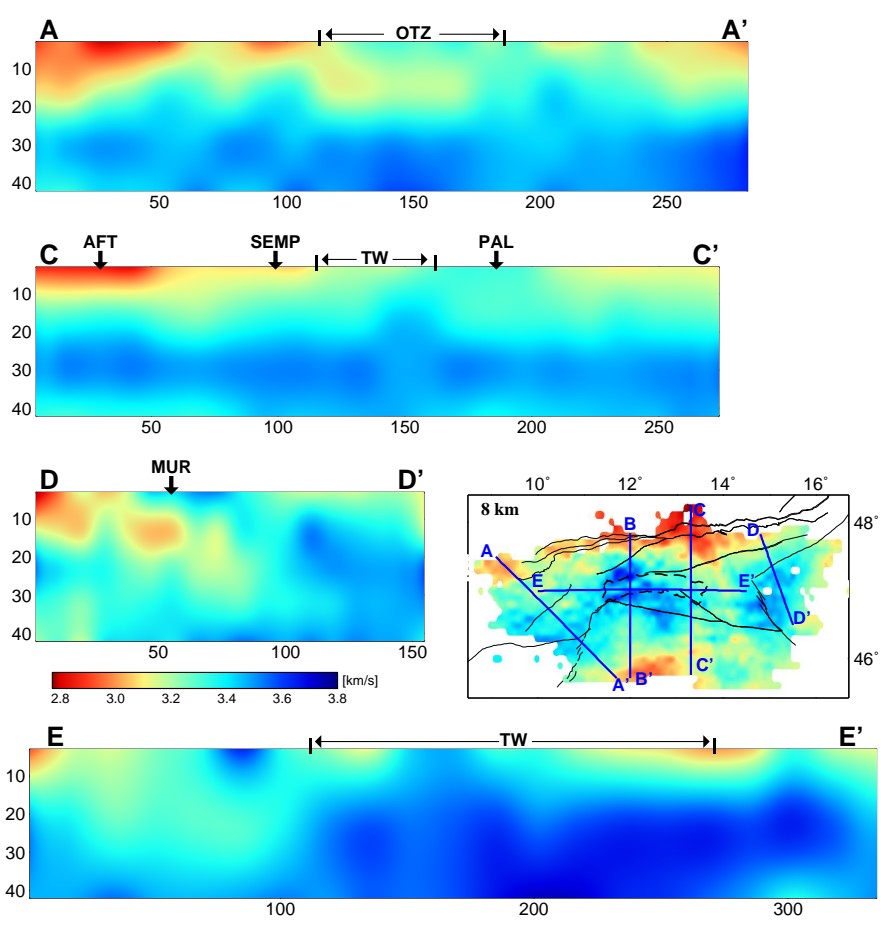

**Figure 14.** Cross-sections of the Rayleigh-wave shear-velocity model. Surface location of the Ötztal block (OTZ), Tauern Window (TW), Periadriatic fault (PAL), Salzach–Ennstal–Mariazell–Puchberg (SEMP), and Mur-Mürz fault (MUR) are shown on the profiles. Profile locations are presented in the 8 km depth slice. The Tauern Window (TW) and the main faults are shown by black lines (see Fig. 1 for faults name).

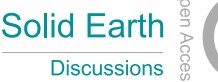

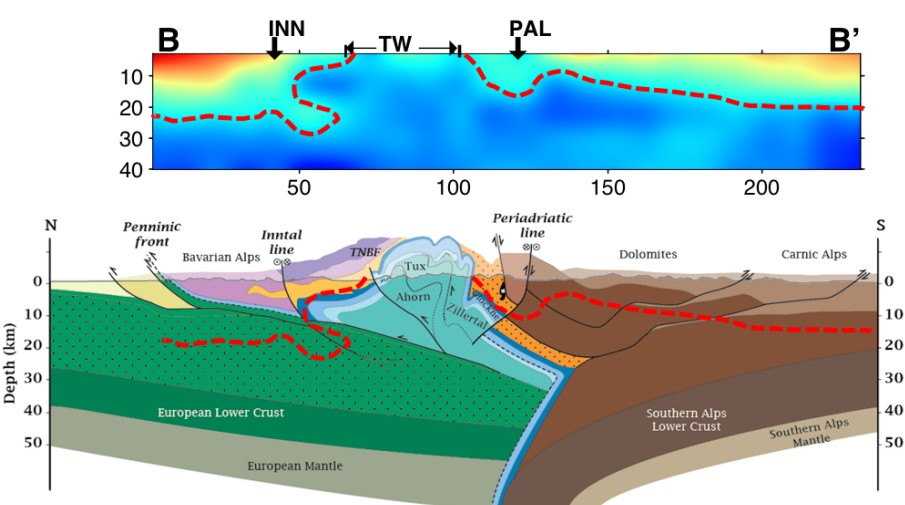

**Figure 15.** Top: cross-section of the RVs model along the TRANSALP profile. Dashed red line separates high-velocity zone associated with the sub-Tauern basement from lower velocity crustal rocks. Bottom: section showing geological interpretation of the TRANSALP in the Surface location of the Tauern (Schmid et al. (2004), see Bousquet et al. (2008)). The same dashed line is superimposed on the geological interpretation of the TRANSALP. Tauern Window (TW), Periadriatic fault (PAL), Inntal line (INN) are shown on the profiles.