# Peer review of "Crustal structures beneath the Eastern and Southern Alps from ambient noise tomography"

_Solid Earth, 2019_

## Referee Comment (RC1) · Anonymous Referee #1 · 6 Mar 2020

This is an interesting study that presents a crustal shear wave velocity model of the eastern and southern Alps by inverting surface wave dispersion data from ambient noise. The final model has clear geological coherence and shows a good correlation between the velocity contrasts and geological structure. But there is still some work to be done. The authors need to perform evaluations of both tomography results and the inversion model to avoid the overinterpretation. I also have concerns about both figures and the manuscript. I hope that the authors will find these suggestions helpful.

Comments:

1) Since the cross-correlation database has been constructed, I am curious why the

phase velocity information is not included in the tomography.

2) The tomography and final inversion results reflect lots of small-scale anomalies and artifacts, which indicates the tomographic results are not robust. I suggest the authors trying to adjust the smoothing parameter and correlation length, during the phase velocity map construction.

3) The synthetic reconstruction analysis with synthetic models is useful to assess the relative spatial resolution, since the ray path coverage is not uniform, especially after the introduction of AlpArray. Even though the synthetic tests, such as the checkboard test, cannot indicate the range of resolvable scale-lengths, it still could reflect the noise sensitivity and parameterization sensitivity. (Rawlinson and Spakman, 2016)

4) The data coverage is bad for the boundary region and in the long period. It's hard to convince me that the anomalies around the boundary and at deep depths (such as I, II, III, V, and X in Figs 11 and 12; high Vsv anomaly in Figs. 11f and 11g; profiles AA and DD in Fig 14) are realistic. So, I suggest the authors avoiding overinterpreting these features.

5) Several figures are not decent, such as Fig. 3, 5, 6, 7, and 8. In addition, the font-size of the labels and titles is too small in some figures.

Specific comments:

2.3:

Figure 3: I do not think the Figure 3 of the 9 components correlation tensor is necessary. The figure is obscure and hard to distinguish the signal from the background.

3:

Figure 4: Typically, we concern more about the period dependent SNR during FTAN, which more related to the quality of the dispersion measurement, rather than the SNR of CC. The SNR in Figure 4 is meaningless. I suggest adding the figure showing the

period dependent SNR curves for both Rayleigh and Love wave (similar to Fig. 4 in Bensen et al. 2008). The analysis of variation of the SNR with inter-station distance and azimuth could be considered at the fixed period.

Line 161: The figure of the period dependent number of the Rayleigh and Love group velocity measurements is necessary, which could be added in the manuscript or supplement. (Similar to Lin et al. 2008)

Figure 5: The example of FTAN seems not to be very well. The maximum period traced in the FTAN example in Fig. 5a and 5b is 30 s, while the period used in the FTAN is from 1 to 50 s. Is this already belong to the best results of FTAN? In addition, the locations of the station pair used in Fig. 5a and 5b could be marked in Fig. 5c.

4:

Line 175: Why is the grid size set as 8 km? How does the grid size affect the tomography results in different periods?

Line 177: May need to consider to use longer correlation lengths.

4.1:

Figure 6: Path density maps only show the periods of 8 and 16 s. What does the path density map look like for the longer periods, such as 30, 40, and 50 s?

Line 191: The resolution length is only the reflection of the relative path density and choice of parameters in the tomography. It does not indicate the true resolution.

Figure 7: The true resolution actually cannot be reflected by the resolution length map, which is also controlled by the model parameterization. This figure is a little bit redundant. I suggest removing it or put into the supplement.

How about the average misfit of the tomography result for different periods? Could you please provide a figure to show the period dependent misfit variation for both Rayleigh and Love wave group velocity?

4.2:

Figure 8: The number of the path in 20 s should be good, but the tomographic result seems not stable. Why are there so many white blanks in Fig. 8d? Could you please provide the Rayleigh wave group velocity map at 30, 40, and 50 s? Besides, the region of the CZA could be labeled in the figure. The full name of SLA should be indicated in the caption.

5:

Line 231: The final group velocity data used in the inversion is from 4 to 50 s, which should be clarified. Why do you exclude the periods from 1 to 4 s in the step of the inversion rather than in performing FTAN? I suggest to cut off the period range during performing FTAN.

Line 237: How do you determine the thickness of the layer in depth?

The influence of Moho depth is not mentioned in the paper. What is the Moho depth distribution in this region? Will the Moho depth affect the inversion? How are other model parameters assigned in the parameterization?

Figure 10: The figure showing a comparison of depth sensitivity kernel of Rayleigh and Love wave group velocity at different periods to Vp and Vs is helpful. I suggest removing the Fig. 10c and 10d and add another figure of the comparison of the depth sensitivity kernel of Rayleigh and Love wave.

6:

Figures 11 and 12. The tectonic boundaries (dashed lines) in Fig. 11 and 12 are not clear and hard distinguished from faults. It will be helpful to label the abbreviations of the tectonic units mentioned in the paper.

Line 346: Another reason for the discrepancies in the pattern of anomalies between your model and Kastle's is the different station distribution. The introduction of AlpArray

stations increases the paths in the central region.

Figure 13. Mark anomaly IX in Fig. 13d. Also, it will be helpful to label the tectonic abbreviations mentioned in Fig. 13a.

Additional reference:

Bensen, G.D., Ritzwoller, M.H. and Shapiro, N.M., 2008. Broadband ambient noise surface wave tomography across the United States. Journal of Geophysical Research: Solid Earth, 113(B5).

Lin, Fan-Chi, Morgan P. Moschetti, and Michael H. Ritzwoller. "Surface wave tomography of the western United States from ambient seismic noise: Rayleigh and Love wave phase velocity maps." Geophysical Journal International 173.1 (2008): 281-298

Rawlinson, N. and Spakman, W., 2016. On the use of sensitivity tests in seismic tomography. Geophysical Journal International, 205(2), pp.1221-1243.

---

## Referee Comment (RC2) · Andreas Fichtner (Referee) · 8 Apr 2020

Dear editor, Dear authors,

First of all, I need to apologise for this very late review! The past few weeks have been rather chaotic, and the transition to online teaching has consumed a lot of time.

This manuscript presents an ambient-noise tomography study of the Eastern and Southern Alps, largely using data that were collected specifically to study Alpine structure. Following a summary of Alpine geology and previous tomographic studies in that region, the authors provide details on data processing, dispersion measurements,

and the two-stage inversion procedure (via group-velocity maps to a 3-D model). The manuscript ends with a detailed discussion of the results in the context of the regional geology.

This contribution is really a pleasure to read because the topic is interesting, and because the work is described carefully without being overly verbose. Not being an expert in geology, I will mostly comment on technical issues. In addition to the points raised below, you can find more suggestions and questions in the annotated manuscript.

Major comments

1) Resolution: My major concern is the resolution analysis. In section 4.1, the authors claim, for instance, that resolution length at 16 s is as low as 8 km. Obviously, this is physically impossible. At 16 s, surface wave velocity is around 3 km/s. Therefore, the wavelength is certainly larger than 50 km. It follows that resolution in this transmission tomography can be at most 50 km at 16 s period.

The problem here seems to be that the authors forget the limitations of ray theory. By virtue of the central-slice theorem, ray theory can give infinite resolution, regardless of the frequency content of the waves. In other words, this apparently good resolution is really just an artefact of the ray approximation.

Another problem is that resolution length is a quantity that has a direction. Resolution in one direction is generally different from resolution in another one. So, which direction do you consider here?

2) Details of the inversion: Some technical details of the inversion procedure could be described better. Especially in the first paragraph of section 4, the authors introduce various parameters that seemly control the regularisation of the inverse problem. Without showing an equation, it is difficult to understand what exactly these parameters are, and how their specific values have been determined.

Minor comments

3) English: The English of the manuscript is good, but can still be improved. For instance, many plural s's are missing. So, I would suggest that a native speaker carefully reads the text.

4) Figures: Some of the figures could be improved. Often, the labels are too small and resolution is a bit low.

5) Others: Please find more smaller comments and questions in the annotated manuscript.

In summary, I think that the resolution analysis needs a little bit more attention. In case the numbers change, the interpretation may need to be adjusted. All in all, this should not require more than a minor revision.

With kind regards

Andreas Fichtner

Please also note the supplement to this comment:
https://www.solid-earth-discuss.net/se-2019-177/se-2019-177-RC2-supplement.pdf

[Figure]

**Supplement:**

[revised manuscript text omitted]

---

## Author Comment (AC1) · 8 Jun 2020

Dear Editor,

We are pleased to submit the authors' response of "Crustal structures beneath the Eastern and Southern Alps from ambient noise tomography". We appreciate the time and attention by the editor, associate editor, and referees. The comments and questions were insightful and enabled us to improve the quality of the manuscript.
All points raised by the reviewer1 have been addressed. In the following we list the reviewer's comments in bold face following by the authors' response to each of

comments and questions. As suggested by the reviewer, we have completely redone the group velocity inversions using larger smoothing parameters. In turn, we have updated 10 Figures (Fig. 6 to Fig. 15). We have also included new Figures and replaced Figure 4 (see details below).

Sincerely,
Ehsan Qorbani on behalf of the co-authors

**Referee 1**

**1-Why is the phase velocity information not included in the tomography?**

In the present study we decided to process only group velocity maps because of the relatively high computational cost of inversions. Indeed, we compute for the whole resolution matrix which increases significantly the computation time for a single period.

**2-The tomography and final inversion results reflect lots of small-scale anomalies and artifacts, which indicates the tomographic results are not robust. I suggest the authors trying to adjust the smoothing parameter and correlation length, during the phase velocity map construction.**

In the initial version of the final inversion, there were indeed some small-scale anomalies that were mainly produced by the interpolation used to produce the maps. In addition, some small-scale features were small artifacts produced by low path density in some regions. Note however that all the main bodies identified and discussed in the manuscript were well defined. But we generally agree with the reviewer that the final maps displayed were perturbed by small scale problems.

In order to solve those issues, we completely remade the group velocity inversions

which resulted in significant changes in the manuscript (figures from 6 to 15 have been updated). Several changes have been made to improve the inversion results. First, as the goal of the work is to study large scale bodies in the crust, we increased the grid size from 8km to 12km. This significantly increases the path density in the whole study region. Secondly, as suggested by the reviewer, we adjusted the inversion parameters using L-curves analysis that are now provided as a new figure in the manuscript. Finally, we carefully check the interpolation used to plot the final images. The new inversion is now smooth with clear marked bodies that are discussed in the geological interpretation section.

**3-The synthetic reconstruction analysis with synthetic models is useful to assess the relative spatial resolution, since the ray path coverage is not uniform, especially after the introduction of AlpArray. Even though the synthetic tests, such as the checkboard test, cannot indicate the range of resolvable scalelengths, it still could reflect the noise sensitivity and parameterization sensitivity. (Rawlinson and Spakman, 2016)**

We do not include a checkerboard test in our study, because we agree with Lévêque et al. 1993 (reference see below) that these tests can be misleading mainly due to the arbitrary choice of the synthetic models to be tested. Assessing the resolution directly from the resolution matrix, as done in the manuscript (see section 4.1), is a more robust way of quantifying resolution (e.g., Barmin et al., 2001, 2012) and the increased computational and storage cost associated with this matrix is manageable. With the Barmin et al. (2001) method, each row of the resolution matrix is a map representing the resolution for one cell of the model. It quantifies how the obtained group velocity at one node depends on the measurements performed at other nodes. This matrix allows to simply define a correlation length as the distance at which the value in the resolution matrix is decreased to half (Barmin et al., 2001; Stehly et al., 2009). Studies combining both resolution matrix analysis and checkerboard tests show similar results

(e.g., Poli et al., 2013) with more information for the resolution matrix approach (e.g., Barmin et al., 2001; Stehly et al., 2009). In particular, as the spatial projection of the individual resolution matrices for each cell are not symmetric, this analysis allows to look at the different size of the resolution spot in the best and worst direction for each cell (we now included these figures to Fig. 6). As a result, we believe that the analysis presented in section 4.1 is the best way to assess the quality of our model.

- Lévêque, J.-J., Rivera, L. and Wittlinger, G. (1993). On the use of the checkerboard test to assess the resolution of tomographic inversions. Geophys. J. Int. 313–318.

**4-The data coverage is bad for the boundary region and in the long period. It's hard to convince me that the anomalies around the boundary and at deep depths (such as I, II, III, V, and X in Figs 11 and 12; high Vsv anomaly in Figs. 11f and 11g; profiles AA and DD in Fig 14) are realistic. So, I suggest the authors avoiding overinterpreting these features.**

In the new version of the manuscript we increased the grid size in order to improve the path density in the whole study region. Therefore, the new images are now better constrained regarding the discussed features. In addition, we include sentences when it's necessary to remind the reader that some features have to be interpreted carefully in regions with low path coverage.

**5- Several figures are not decent, such as Fig. 3, 5, 6, 7, and 8. In addition, the font-size of the labels and titles is too small in some figures.**

We improve the quality of the Figures and modify the titles and labels in the revised manuscript.

Specific comments

Section 2.3:

**Figure 3: I do not think the Figure 3 of the 9 components correlation tensor is necessary. The figure is obscure and hard to distinguish the signal from the background.**

We agree with the reviewer and move the Figure 3 to Supplementary materials.

Section 3:

**Figure 4: Typically, we concern more about the period dependent SNR during FTAN, which more related to the quality of the dispersion measurement, rather than the SNR of CC. The SNR in Figure 4 is meaningless. I suggest adding the figure showing the period dependent SNR curves for both Rayleigh and Love wave (similar to Fig. 4 in Bensen et al. 2008). The analysis of variation of the SNR with inter-station distance and azimuth could be considered at the fixed period. Line 161: The figure of the period dependent number of the Rayleigh and Love group velocity measurements is necessary, which could be added in the manuscript or supplement. (Similar to Lin et al. 2008)**

The authors thank the reviewer for the suggestion. We replace the Figure 4 by period dependence SNR of the dispersion measurements. The figure shows variation of average SNR versus period for Rayleigh-wave and the four inter-components, ZZ, ZR, RR, RZ, which the Rayleigh-waves are constructed from. Also for Love-wave, The TT inter-component that Love-waves are appears on (Figure AC1 is attached to this letter). In the revised version, we add a supplementary table including number of measurements for each period for Rayleigh and Love waves, selected after applying a number of criteria explained section 3 of manuscript.

**Figure 5: The example of FTAN seems not to be very well. The maximum period traced in the FTAN example in Fig. 5a and 5b is 30 s, while the period used in the FTAN is from 1 to 50 s. Is this already belong to the best results of FTAN? In addition, the locations of the station pair used in Fig. 5a and 5b could be marked in Fig. 5c.**

The example of FTAN in Figure 5 have been randomly chosen. The max 25 and 30 sec in those examples are dependent on their inter-station distances. We now add more examples of FTAN in the supplementary figures.

Section 4:
**Line 175: Why is the grid size set as 8 km? How does the grid size affect the tomography results in different periods?**

We initially selected 8km as it was the smallest grid size that was still allowing sufficient number of paths per cell in most of the study region. However, in the new version of the manuscript we decided to increase the grid-size to 12km. This increases the number of measurements in each cell over the whole study region and help to stabilize the inversions.

**Line 177: May need to consider to use longer correlation lengths.**

As explained in major point number 2, all the parameters including grid size, correlation length (alpha) and damping parameters (sigma) have been changed in the new version of the manuscript.

Section 4.1:
**Figure 6: Path density maps only show the periods of 8 and 16 s. What does the path density map look like for the longer periods, such as 30, 40, and 50 s?**

We have redone the inversions with larger grid size (12 km) and smoothing parameters. Path density for longer periods for instance at the 20s, 30s, and 40s period is good enough to resolve the structures. The average number of paths per cell for Rayleigh-wave are 28, 23, 22 for 20s, 30s, and 40s respectively; and for Love-wave are 29, 26, 25 for 20s, 30s, and 40s. Please see attached Figure AC2 to the author's response.

**Line 191: The resolution length is only the reflection of the relative path density and choice of parameters in the tomography. It does not indicate the true resolution.**

This is true. We therefore have changed the terminology used in the manuscript to better reflect that this measurement is simply a proxy to assess the spatial averaging of the inversion rather than the true resolution of the model. "Resolution length" has been renamed "correlation length", which better explain that this value may be interpreted as the minimum distance at which two delta-shaped input anomalies can be resolved on the tomographic map (Barmin et al., 2001). However, we would like to point that assessing the "resolution" in the sense of "correlation length" directly from the resolution matrix, as done in the manuscript, is a robust way of quantifying spatial averaging and the size of the "resolution" spots (Barmin et al., 2001; An 2012). Studies combining both resolution matrix analysis and checkerboard tests show similar results in terms of extracted correlation length (e.g., Poli et al., 2013) with more information for the resolution matrix approach (e.g., Barmin et al., 2001; Stehly et al., 2009). In particular, as the spatial projection of the individual resolution matrices for each cell are not symmetric, this analysis allows to look at the different size of the "resolution spot" in the best and worst direction for each cell. We added a few sentences in the text for better explanation.

**Figure 7: The true resolution actually cannot be reflected by the resolution length map, which is also controlled by the model parameterization. This figure is a little**

**bit redundant. I suggest removing it or put into the supplement.**

We prefer to keep this figure in the main text as it better reflects the spatial averaging of the model than the number of paths per cell. This information is relevant as it provides an idea of size over which the inversions averaged the measurements to produce the model. It is therefore useful to interpret the models.

**How about the average misfit of the tomography result for different periods? Could you please provide a figure to show the period dependent misfit variation for both Rayleigh and Love wave group velocity?**

In inversion procedures, in general, a search is performed to find the best values of model parameters, which minimize the misfit or variance. Our standard way of selecting the optimum set of parameters (damping factor and correlation length) in the group velocity inversion is to evaluate how much the model reduces the variance present in data. In response to the reviewer's comment, we included graphs of variance reduction changes for several selections of the two parameters, damping factor (alpha) and correlation length (sigma) for a selection of periods. We have also attached this figure (Fig. AC3) to the author's response. In the figure, our selected best parameters are shown in black circles.

Section 4.2:
**Figure 8: The number of the path in 20 s should be good, but the tomographic result seems not stable. Why are there so many white blanks in Fig. 8d?**

As mentioned in previous comments, we have changed all the figures from 6 to 15. The new models are more stable. The white blanks in the old version were simply cells for which the number of paths was not sufficient (below 5). The cell exists in the model but were simply not plotted because of the low path density.

**Could you please provide the Rayleigh wave group velocity map at 30, 40, and 50 s? Besides, the region of the CZA could be labeled in the figure. The full name of SLA should be indicated in the caption.**

In the study, the group velocities between 4s and 42s have been used for the inversion of shear-velocities. We provide Rayleigh-wave group velocity maps at 30s and 40s and include them into supplementary materials. We add the full name of SLA to the caption of the Figure 8 and Figure 9.

Section 5:

**Line 231: The final group velocity data used in the inversion is from 4 to 50 s, which should be clarified. Why do you exclude the periods from 1 to 4 s in the step of the inversion rather than in performing FTAN? I suggest to cut off the period range during performing FTAN.**

We performed FTAN for all period range that we had. We then carefully assessed the dispersion measurements by apply a number of criteria, filter them, and finally to select the period range that are well constrained to be used in the group velocity inversion. We found out that measurements below 4 sec might not represent group velocity of the fundamental mode and could be mistaken by higher modes. Therefore, we did not include periods less than 4s in to the inversion. We also did not use period larger than 42s; because of lower number of measurements, they might not have been well constrained. We now add sentences to manuscript for better explanation.

**Line 237: How do you determine the thickness of the layer in depth?**

The initial model for the shear-velocity inversion is coming from Behm et al. (2007a) with depth layering of 1 km. From that model we obtained our an average initial model for the region with 2km layers in depth. Since we cut off the group velocities below 4 sec, the thickness of the layers was increased from 1 km to 2 km. The choice was

made as it allows a good enough discretization of the 1D shear-velocity model with depth while limiting the number of parameters in the inversion.
**The influence of Moho depth is not mentioned in the paper. What is the Moho depth distribution in this region? Will the Moho depth affect the inversion? How are other model parameters assigned in the parameterization?**

The Moho depth in the region has been reported > 40 km (Behm et al., 2007a; Spada et al., 2013; Bianchi et al., 2015, Hetenyi et al., 2018). Our shear velocity model presented here ends at 40 km, therefore we are not able to observe effect of Moho in our model. During the depth inversion, no restriction was applied for Moho depth, and also no layer weighting, and no fixed velocity was set. The velocity is allowed to take a large range of values as long as the depth variation is smooth.

**Figure 10: The figure showing a comparison of depth sensitivity kernel of Rayleigh and Love wave group velocity at different periods to Vp and Vs is helpful. I suggest removing the Fig. 10c and 10d and add another figure of the comparison of the depth sensitivity kernel of Rayleigh and Love wave.**

We added a figure showing the depth sensitivity kernel of Rayleigh and Love waves to Vs at different periods (see Figure 10 in the updated version). We prefer to keep Fig. 10c and 10d as they provided useful information on the depth resolution of the Vs inversions.

Section 6:
**Figures 11 and 12. The tectonic boundaries (dashed lines) in Fig. 11 and 12 are not clear and hard distinguished from faults. It will be helpful to label the abbreviations of the tectonic units mentioned in the paper. Figure 13: Mark anomaly IX in Fig. 13d. Also, it will be helpful to label the tectonic abbreviations mentioned**

[Figure]

**in Fig. 13a.**

Having both tectonic boundaries and faults would be helpful for readers to follow the interpretation. We add the label of tectonic units to Figure 11, 12, and 13a, and mark anomaly IX to Figure 13d.

**Line 346: Another reason for the discrepancies in the pattern of anomalies between your model and Kastle's is the different station distribution. The introduction of AlpArray stations increases the paths in the central region.**

We thank the reviewer for comments and suggestions we include this to manuscript.

**Figures caption**

Figure AC1: Average signal-to-noise ratio (SNR) for Rayleigh and Love waves for all station pairs. Average SNR of ZZ, RR, ZR, and RZ are also shown, which Rayleigh waves are extracted from these 4 inter-components. Average SNR of TT and Love-waves are also represented in the figure. Love waves appears on the TT inter-components.

Figure AC2: Path density map for the group velocity inversion: a, b, c) for Rayleigh waves at 20, 30, and 40 sec respectively. d, e, f) for Love waves at 20, 30, and 40 sec. The path coverage is generally good for the entire region.

Figure AC3: Variance reduction as a function of the inversion parameters. a) L-curve analysis for damping factor (alpha) for Rayleigh waves at periods of 5, 10, 20, 30, 40 sec. b) Correlation length (sigma) for Rayleigh waves at the same period range. c,

d) Variance reduction vs damping factor and correlation length (sigma) respectively for Love waves at the same period range. The selected parameters are shown by black circles.

---

## Author Comment (AC2) · 8 Jun 2020

Dear Editor,

We are pleased to submit the authors' response of "Crustal structures beneath the Eastern and Southern Alps from ambient noise tomography". We appreciate the time and attention by the editor, associate editor, and referees. The comments and questions were insightful and enabled us to improve the quality of the manuscript.
All points raised by the reviewer2 have been addressed. In the following we list the reviewer's comments in bold face following by the authors' response to each of

comments and questions. As suggested by the reviewer, we modified the text to better explain of the resolution of the model. We updated Figure 6 (of the revised manuscript) to include correlation length in best and the worst directions as well. We also add more equations and explanations to the text to better describe details of the group velocity inversion.

Sincerely,
Ehsan Qorbani on behalf of the co-authors

**Referee 2**

**1) Resolution: My major concern is the resolution analysis. In section 4.1, the authors claim, for instance, that resolution length at 16 s is as low as 8 km. Obviously, this is physically impossible. At 16 s, surface wave velocity is around 3 km/s. Therefore, the wavelength is certainly larger than 50 km. It follows that resolution in this transmission tomography can be at most 50 km at 16 s period. The problem here seems to be that the authors forget the limitations of ray theory. By virtue of the central-slice theorem, ray theory can give infinite resolution, regardless of the frequency content of the waves. In other words, this apparently good resolution is really just an artefact of the ray approximation. Another problem is that resolution length is a quantity that has a direction. Resolution in one direction is generally different from resolution in another one. So, which direction do you consider here?**

We thank the reviewer for its comment and we agree with him. The resolution as discussed in the manuscript is indeed not a true resolution but more an estimation of the correlation length or the size of the averaging spot for each cell of the model. We also agreed with the reviewer that the text was misleading, and we therefore have changed

the terminology used in the manuscript to better reflect that this measurement is simply a proxy to assess the spatial averaging of the inversion rather than the true resolution of the model. "Resolution length" has been renamed "correlation length", which better explain that this value may be interpreted as the minimum distance at which two delta-shaped input anomalies can be resolved on the tomographic map (Barmin et al., 2001). We would like to point that assessing the "resolution" in the sense of "correlation length" directly from the resolution matrix, as done in the manuscript, is a robust way of quantifying spatial averaging and the size of the "resolution" spots (Barmin et al., 2001; An 2012). Studies combining both resolution matrix analysis and checkerboard tests show similar results in terms of extracted correlation length (e.g., Poli et al., 2013) with more information for the resolution matrix approach (e.g., Barmin et al., 2001; Stehly et al., 2009). As mentioned by the reviewer, the spatial projection of the individual resolution matrices for each cell are not symmetric which allows to look at the different size of the "resolution spot" in the best and worst direction for each cell. In the previous version of the paper, we presented the mean "correlation length" between the best and worst direction. In the updated version of the paper we now include the mean correlation length, the one in the best direction and the one in the worst direction (Fig. 6 in the revised version).

**2) Details of the inversion: Some technical details of the inversion procedure could be described better. Especially in the first paragraph of section 4, the authors introduce various parameters that seemly control the regularisation of the inverse problem. Without showing an equation, it is difficult to understand what exactly these parameters are, and how their specific values have been determined.**

In the revised manuscript, we add more explanation and include equations for more clarification.

**3) English: The English of the manuscript is good, but can still be improved. For instance, many plural s's are missing. So, I would suggest that a native speaker carefully reads the text.**

We will ask a native English speaker to proofread the revised manuscript.

**4) Figures: Some of the figures could be improved. Often, the labels are too small and resolution is a bit low.**

We improve the quality of the figures and make the labels and titles larger.

**The authors' response to the major comments in the supplement by Referee2**

**1-It is not quite clear why you would need 9. Already the 3 diagonal components would allow you to do this.**

To compute the diagonal terms ZZ, TT, and RR of the correlation tensor, the rotation matrix includes ZZ, EE, EN, NE and NN terms. In addition, the RZ and ZR terms also includes Rayleigh waves. Calculating the full tensor therefore allows redundancy in the dispersion measurements which avoid biases due to noise sources distribution.

**2-"*In order to select the clearer CC, we picked those that have signal-to-noise ratio (SNR) larger than 4*". This seems to be a very low signal-to-noise ratio.**

The selection based on SNR is done at the very beginning of the selection process. At that step, the database is still composed in most of the correlation pairs. This low SNR is simply there to remove the really poor correlation functions. Most of the actual selection is performed by the following selection criteria. To better show the SNR of the

final correlations, we added a new figure (Fig. 3 in the revised manuscript) presenting the average SNR as a function of period. This figure clearly shows that the SNR of the correlation entering the inversion is actually good for all periods. The figure is also attached to this letter (see Fig AC1).

**3- "***To improve the reliability of Rayleigh wave dispersions measurements we used the redundancy of the correlation tensor by using all components (RR, RZ, ZR, and ZZ) containing Rayleigh waves***". It is not clear that this really brings an improvement. In fact, you silently make the assumption that the Earth is isotropic. In case of azimuthal anisotropy, which surely exists but may be difficult to constrain, you introduce a systematic error, e.g., by having Love waves on the vertical component and vice versa.**

The goal of the paper is to present a isotropic velocity model of the Eastern Alps. The combination of the various components containing Rayleigh waves improves the stability of the dispersions measurements by limiting the bias due to the non-uniform noise sources distribution. This type of combination is commonly done in Ambient Noise Tomography to improve the stability of the dispersions measurements.

**4-"***These parameters strongly affect the variance reduction of the final model. Stehly et al. (2009) recommended that the correlation length should be at least equal to grid size***". Since you explicitly say that these parameters affect the variance reduction, it is difficult to understand that you just fix them. How did you get to these values? For instance, did you run some kind of L-curve analysis for them?**

We selected the inversion best parameters using L-curves analysis . The variations of variance reduction as a function of damping factor and correlation length that are now provided as a new figure in the manuscript. The Figure is also attached to this letter

(Figure AC3).

**Figures caption**

Figure AC1: Average signal-to-noise ratio (SNR) for Rayleigh and Love waves for all station pairs. Average SNR of ZZ, RR, ZR, and RZ are also shown, which Rayleigh waves are extracted from these 4 inter-components. Average SNR of TT and Love-waves are also represented in the figure. Love waves appears on the TT inter-components.

Figure AC3: Variance reduction as a function of the inversion parameters. a) L-curve analysis for damping factor (alpha) for Rayleigh waves at periods of 5, 10, 20, 30, 40 sec. b) Correlation length (sigma) for Rayleigh waves at the same period range. c, d) Variance reduction vs damping factor and correlation length (sigma) respectively for Love waves at the same period range. The selected parameters are shown by black circles.

---

## Author Response (AR1)

Dear Editor,

We are pleased to submit the response to referee's comments of "Crustal structures beneath the Eastern and Southern Alps from ambient noise tomography". We would like to thank the editor and reviewers for their time spent on reviewing the manuscript. The comments and questions were insightful and helped us to improve the quality of the manuscript.

All points raised by the reviewers have been carefully taken under consideration and addressed. In the following we list the reviewers' comments in bold face following by the authors' response to each of comments and questions and the author's changes in manuscript. As suggested by the reviewer 1, we have completely redone the group velocity inversions using larger smoothing parameters. We have updated all figures and also included new Figures into the manuscript and to the supplementary materials (see details below). This letter is followed by a marked-up version of the revised manuscript in which all changes in the text have been highlighted.

Sincerely,
Ehsan Qorbani on behalf of the co-authors

**Referee 1**

**Comment 1) Why is the phase velocity information not included in the tomography?**

Response 1) In the present study we decided to process only group velocity maps because of the relatively high computational cost of inversions. Indeed, we computed for the whole resolution matrix which increases significantly the computation time for a single period.

**Comment 2) The tomography and final inversion results reflect lots of small-scale anomalies and artifacts, which indicates the tomographic results are not robust. I suggest the authors trying to adjust the smoothing parameter and correlation length, during the phase velocity map construction.**

Response 2) In the initial version of manuscript, there were indeed some small-scale anomalies that were mainly produced by the interpolation used to produce the maps. In addition, some small-scale features were minor artifacts produced by low path density in some regions. Note however that all the main bodies identified and discussed in the manuscript were well defined. But we generally agree with the reviewer that the final maps displayed were perturbed by small scale problems.

In order to solve those issues, we completely remade the group velocity inversions which resulted in significant changes in the manuscript (figures from 6 to 15 have been updated). Several changes have been made to improve the inversion results. First, as the goal of the work is to study large scale bodies in the crust, we increased the grid size from 8km to 12km. This significantly increases the path density in the whole study region. Secondly, as suggested by the reviewer, we adjusted the inversion parameters using L-curves analysis that are now provided as a new Figure 5 in the manuscript. Finally, we carefully check the interpolation used to plot the final images. The new inversion is now smooth with clear marked bodies that are discussed in the geological interpretation section.

**Comment 3) The synthetic reconstruction analysis with synthetic models is useful to assess the relative spatial resolution, since the ray path coverage is not uniform, especially after the introduction of AlpArray. Even though the synthetic tests, such as the checkboard test, cannot indicate the range of resolvable scale-lengths, it still could reflect the noise sensitivity and parameterization sensitivity. (Rawlinson and Spakman, 2016)**

Response 3) We do not include a checkerboard test in our study, because we agree with Lvque et al. 1993 (reference see below) that these tests can be misleading mainly due to the arbitrary choice of the synthetic models to be tested. Assessing the resolution directly from the resolution matrix, as done in the manuscript (see section 4.1), is a more robust way of quantifying resolution (e.g., Barmin et al., 2001, 2012) and the increased computational and storage cost associated with this matrix is manageable. With the Barmin et al. (2001) method, each row of the resolution matrix is a map representing the resolution for one cell of the model. It quantifies how the obtained group velocity at one node depends on the measurements performed at other nodes. This matrix allows to simply define a correlation length as the distance at

which the value in the resolution matrix is decreased to half (Barmin et al., 2001; Stehly et al., 2009). Studies combining both resolution matrix analysis and checkerboard tests show similar results (e.g., Poli et al., 2013) with more information for the resolution matrix approach (e.g., Barmin et al., 2001; Stehly et al., 2009). In particular, as the spatial projection of the individual resolution matrices for each cell are not symmetric, this analysis allows to look at the different size of the resolution spot in the best and worst direction for each cell (we now included these figures to Fig. 6). As a result, we believe that the analysis presented in section 4.1 is the best way to assess the quality of our model.

- Lvque, J.-J., Rivera, L. and Wittlinger, G. (1993). On the use of the checkerboard test to assess the resolution of tomographic inversions. Geophys. J. Int. 313318.

**Comment 4) The data coverage is bad for the boundary region and in the long period. It's hard to convince me that the anomalies around the boundary and at deep depths (such as I, II, III, V, and X in Figs 11 and 12; high Vsv anomaly in Figs. 11f and 11g; profiles AA and DD in Fig 14) are realistic. So, I suggest the authors avoiding overinterpreting these features.**

Response 4) In the new version of the manuscript we have increased the grid size in order to improve the path density in the whole study region. Therefore, the new images are now better constrained regarding the discussed features. In addition, we have included sentences when it is necessary to remind the reader that some features have to be interpreted carefully in regions with low path coverage (e.g. Lines 282-285, 290-292). We have modified and ignored parts of the manuscript in order to avoid over interpretation: Lines 293-297, 314-315, 326-329 was removed from the original manuscript.

**Comment 5) Several figures are not decent, such as Fig. 3, 5, 6, 7, and 8. In addition, the font-size of the labels and titles is too small in some figures.**

Response 5) We have updated all figures in the manuscript and have improved the quality of the figures. The titles and labels of figures in the in the revised manuscript were also modifies to be more visible (e.g. Figure 1 and 2).

**Specific comments**

Section 2.3:

**Figure 3: I do not think the Figure 3 of the 9 components correlation tensor is necessary. The figure is obscure and hard to distinguish the signal from the background.**

Response) We agree with the reviewer and have moved the Figure 3 of the original version to Supplementary materials as Figure S1.

Section 3:

**Figure 4: Typically, we concern more about the period dependent SNR during FTAN, which more related to the quality of the dispersion measurement, rather than the SNR of CC. The SNR in Figure 4 is meaningless. I suggest adding the figure showing the period dependent SNR curves for both Rayleigh and Love wave (similar to Fig. 4 in Bensen et al. 2008). The analysis of variation of the SNR with inter-station distance and azimuth could be considered at the fixed period. Line 161: The figure of the period dependent number of the Rayleigh and Love group velocity measurements is necessary, which could be added in the manuscript or supplement. (Similar to Lin et al. 2008)**

Response) The authors thank the reviewer for the suggestion. We have replaced now the Figure 4 by period dependence SNR of the dispersion measurements. The figure shows variation of average SNR versus period for Rayleigh-wave and the four inter-components, ZZ, ZR, RR, RZ, which the Rayleigh-waves are constructed from. Also for Love-wave, The TT inter-component that Love-waves are appears on. In the revised version, we have included a table including number of measurements for each period for Rayleigh and Love waves, selected after applying a number of criteria explained section 3 of manuscript (see supplementary Table S1).

**Figure 5: The example of FTAN seems not to be very well. The maximum period traced in the FTAN example in Fig. 5a and 5b is 30 s, while the period used in the FTAN is from 1 to 50 s. Is this already belong to the best results of FTAN? In addition, the locations of the station pair used in Fig. 5a and 5b could be marked**

**in Fig. 5c.**

Response) The example of FTAN in Figure 5 have been randomly chosen. The max 25 and 30 sec in those examples are dependent on their inter-station distances. We have now added more examples of FTAN in the supplementary Figure S2.

Section 4:
**Line 175: Why is the grid size set as 8 km? How does the grid size affect the tomography results in different periods?**

Response) We initially selected 8 km as it was the smallest grid size that was still allowing sufficient number of paths per cell in most of the study region. However, in the new version of the manuscript we decided to increase the grid-size to 12 km. This increases the number of measurements in each cell over the whole study region and help to stabilize the inversions.

**Line 177: May need to consider to use longer correlation lengths.**

Response) As explained in major point number 2, all the parameters including grid size, correlation length (alpha) and damping parameters (sigma) have been changed in the new version of the manuscript. The inversion parameters selected for each period range are summarized in Supplementary Table S2.

Section 4.1:
**Figure 6: Path density maps only show the periods of 8 and 16 s. What does the path density map look like for the longer periods, such as 30, 40, and 50 s?**

Response) We have redone the inversions with larger grid size (12 km) and smoothing parameters. Path density for longer periods for instance at the 20 s, 30 s, and 40 s period are good enough to resolve the structures. The average number of paths per cell for Rayleigh-wave are 28, 23, 22 for 20 s, 30 s, and 40 s respectively; and for Love-wave are 29, 26, 25 for 20 s, 30 s, and 40 s. We have included figure of path density map at 20 s, 30 s, and 40 s period as Supplementary Figure S3.

**Line 191: The resolution length is only the reflection of the relative path density and choice of parameters in the tomography. It does not indicate the true resolution.**

Response) This is true. We therefore have changed the terminology used in the manuscript to better reflect that this measurement is simply a proxy to assess the spatial averaging of the inversion rather than the true resolution of the model. "Resolution length" has been renamed "correlation length", which better explain that this value may be interpreted as the minimum distance at which two delta-shaped input anomalies can be resolved on the tomographic map (Barmin et al., 2001). However, we would like to point that assessing the "resolution" in the sense of "correlation length" directly from the resolution matrix, as done in the manuscript, is a robust way of quantifying spatial averaging and the size of the "resolution" spots (Barmin et al., 2001; An 2012). Studies combining both resolution matrix analysis and checkerboard tests show similar results in terms of extracted correlation length (e.g., Poli et al., 2013) with more information for the resolution matrix approach (e.g., Barmin et al., 2001; Stehly et al., 2009). In particular, as the spatial projection of the individual resolution matrices for each cell are not symmetric, this analysis allows to look at the different size of the "resolution spot in the best and worst direction for each cell. We added a few sentences in the text for better explanation (section 4.1, lines 181, and lines 187-201).

**Figure 7: The true resolution actually cannot be reflected by the resolution length map, which is also controlled by the model parameterization. This figure is a little bit redundant. I suggest removing it or put into the supplement.**

Response) We prefer to keep this figure in the main text as it better reflects the spatial averaging of the model than the number of paths per cell. This information is relevant as it provides an idea of size over which the inversions averaged the measurements to produce the model. It is therefore useful to interpret the models.

**How about the average misfit of the tomography result for different periods? Could you please provide a figure to show the period dependent misfit variation for both Rayleigh and Love wave group velocity?**

Response) In inversion procedures, in general, a search is performed to find the best values of model parameters, which minimize the misfit or variance. Our standard way of selecting the optimum set of parameters (damping factor and correlation length) in the group velocity inversion is to evaluate how much the model reduces the variance present in data. In response to the reviewer's comment, we included graphs of variance reduction changes for several selections of the two parameters, damping factor (alpha) and correlation length (sigma) for a selection of periods. See Figure 5 of the revised manuscript.

Section 4.2:
**Figure 8: The number of the path in 20 s should be good, but the tomographic result seems not stable. Why are there so many white blanks in Fig. 8d?**

Response) As mentioned in previous comments, we have changed all the figures from 6 to 15. The new models are more stable. The white blanks in the original version were simply cells for which the number of paths was not sufficient (below 5). The cell exists in the model but were simply not plotted because of the low path density.

**Could you please provide the Rayleigh wave group velocity map at 30, 40, and 50 s? Besides, the region of the CZA could be labeled in the figure. The full name of SLA should be indicated in the caption.**

Response) The group velocities between 4 s and 42 s have been used for the inversion of shear-velocities. We have provide Rayleigh-wave group velocity maps at 30 s and 40 s and include them into supplementary materials as Figure S5. We add the full name of SLA to the caption of the Figure 8 and Figure 9 of the revised manuscript.

Section 5:
**Line 231: The final group velocity data used in the inversion is from 4 to 50 s, which should be clarified. Why do you exclude the periods from 1 to 4 s in the step of the inversion rather than in performing FTAN? I suggest to cut off the period range during performing FTAN.**

Response) We performed FTAN for all period range that we had. We then carefully assessed the dispersion measurements by applying a number of criteria, filtering them, and finally to select the period range that are well constrained to be used in the group velocity inversion. We found out that measurements below 4 sec might not represent group velocity of the fundamental mode and could be mistaken by higher modes. Therefore, we did not include periods less than 4 s in to the inversion. We also did not use period larger than 42 s; because of lower number of measurements, they might not have been well constrained. We have now added sentences to manuscript for better explanation (section 5, lines 236-238)

**Line 237: How do you determine the thickness of the layer in depth?**

Response) The initial model for the shear-velocity inversion is coming from Behm et al. (2007a) with depth layering of 1 km. From that model we obtained our an average initial model for the region with 2km layers in depth. Since we cut off the group velocities below 4 sec, the thickness of the layers was increased from 1 km to 2 km. The choice was made as it allows a good enough discretization of the 1D shear-velocity model with depth while limiting the number of parameters in the inversion.

**The influence of Moho depth is not mentioned in the paper. What is the Moho depth distribution in this region? Will the Moho depth affect the inversion? How are other model parameters assigned in the parameterization?**

Response) The Moho depth in the region has been reported >40 km (Behm et al., 2007a; Spada et al., 2013; Bianchi et al., 2015, Hetenyi et al., 2018). Our shear velocity model presented here ends at 40 km, therefore we are not able to observe effect of Moho in our model. During the depth inversion, no restriction was applied for Moho depth, and also no layer weighting, and no fixed velocity was set. The velocity was allowed to take a large range of values as long as the depth variation is smooth.

**Figure 10: The figure showing a comparison of depth sensitivity kernel of Rayleigh and Love wave group velocity at different pe-**

**riods to Vp and Vs is helpful. I suggest removing the Fig. 10c and 10d and add another figure of the comparison of the depth sensitivity kernel of Rayleigh and Love wave.**

Response) We have added the depth sensitivity kernel of Rayleigh and Love waves to Vs at different periods (see lines 252-252 and Figure 10 in the revised version). We prefer to keep the histogram of the distribution of misfit between synthetic and observed dispersion curves as they provided useful information on the depth resolution of the Vs inversions.

Section 6:
**Figures 11 and 12. The tectonic boundaries (dashed lines) in Fig. 11 and 12 are not clear and hard distinguished from faults. It will be helpful to label the abbreviations of the tectonic units mentioned in the paper. Figure 13: Mark anomaly IX in Fig. 13d. Also, it will be helpful to label the tectonic abbreviations mentioned in Fig. 13a.**

Response) Having both tectonic boundaries and faults would be helpful for readers to follow the interpretation. The label of tectonic units are shown and explained in Figure 8 and 9. In addition, we have marked the anomalies discussed in the text as roman number in Figure 11 and 12. We have now marked anomaly VII (previously as XI in the original version) to Figure 13.

**Line 346: Another reason for the discrepancies in the pattern of anomalies between your model and Kastle's is the different station distribution. The introduction of AlpArray stations increases the paths in the central region.**

Response) We thank the reviewer for comments and suggestions we include this to manuscript (line 343-344 of the revised manuscript).

**Referee 2**

**Comment 1) Resolution: My major concern is the resolution analysis. In section 4.1, the authors claim, for instance, that resolution length at 16 s is as low as 8 km. Obviously, this is physically impossible. At 16 s, surface wave velocity is around 3 km/s. Therefore, the wavelength is certainly larger than 50 km. It follows that resolution in this transmission tomography can be at most 50 km at 16 s period. The problem here seems to be that the authors forget the limitations of ray theory. By virtue of the central-slice theorem, ray theory can give infinite resolution, regardless of the frequency content of the waves. In other words, this apparently good resolution is really just an artefact of the ray approximation. Another problem is that resolution length is a quantity that has a direction. Resolution in one direction is generally different from resolution in another one. So, which direction do you consider here?**

Response 1) We thank the reviewer for its comment and we agree with him. The resolution as discussed in the manuscript is indeed not a true resolution but more an estimation of the correlation length or the size of the averaging spot for each cell of the model. We also agreed with the reviewer that the text was misleading, and we therefore have changed the terminology used in the manuscript to better reflect that this measurement is simply a proxy to assess the spatial averaging of the inversion rather than the true resolution of the model. "Resolution length" has been renamed "correlation length", which better explain that this value may be interpreted as the minimum distance at which two delta-shaped input anomalies can be resolved on the tomographic map (Barmin et al., 2001). We would like to point that assessing the "resolution" in the sense of "correlation length" directly from the resolution matrix, as done in the manuscript, is a robust way of quantifying spatial averaging and the size of the "resolution" spots (Barmin et al., 2001; An 2012). Studies combining both resolution matrix analysis and checkerboard tests show similar results in terms of extracted correlation length (e.g., Poli et al., 2013) with more information for the resolution matrix approach (e.g., Barmin et al., 2001; Stehly et al., 2009). As mentioned by the reviewer, the spatial projection of the individual resolution matrices for each cell are not symmetric which allows to look at the different size of the "resolution spot" in the best and worst direction for each cell. In the previous version of

the paper, we presented the mean "correlation length" between the best and worst direction (see section 4.1 in the revised manuscript). In the updated version of the paper we have now included the mean correlation length, the one in the best direction and the one in the worst direction (see Figure 7 in the revised version).

**Comment 2) Details of the inversion: Some technical details of the inversion procedure could be described better. Especially in the first paragraph of section 4, the authors introduce various parameters that seemly control the regularisation of the inverse problem. Without showing an equation, it is difficult to understand what exactly these parameters are, and how their specific values have been determined.**

Response 2) In the revised manuscript, we have now added more explanation and also have included full equations of the inversion method for more clarification. See section 4, lines 157-167

**Comment 3) English: The English of the manuscript is good, but can still be improved. For instance, many plural s's are missing. So, I would suggest that a native speaker carefully reads the text.**

Response 3) We asked a native English speaker to proofread the revised manuscript.

**Comment 4) Figures: Some of the figures could be improved. Often, the labels are too small and resolution is a bit low.**

Response 4) All figures of the revised manuscript have been updated and have been improve the quality. We have also made the labels and titles larger, particularly Figure 1, 2, and 10 that were mentioned by reviewer 2.

**The authors' response to the major comments in the supplement by Referee 2**

In the following we respond to the major comments by the reviewer 2. Other minor comments in the supplement are mainly related to text editing, and have been also implemented to manuscript and are highlighted in the marked-up version of the revised manuscript.

**Comment I) It is not quite clear why you would need 9. Already the 3 diagonal components would allow you to do this.**

Response I) To compute the diagonal terms ZZ, TT, and RR of the correlation tensor, the rotation matrix includes ZZ, EE, EN, NE and NN terms. In addition, the RZ and ZR terms also includes Rayleigh waves. Calculating the full tensor therefore allows redundancy in the dispersion measurements which avoid biases due to noise sources distribution.

**Comment II) "*In order to select the clearer CC, we picked those that have signal-to-noise ratio (SNR) larger than 4*". This seems to be a very low signal-to-noise ratio.**

Response II) The selection based on SNR is done at the very beginning of the selection process. At that step, the database is still composed in most of the correlation pairs. This low SNR is simply there to remove the really poor correlation functions. Most of the actual selection is performed by the following selection criteria. To better show the SNR of the final correlations, we have added a new figure (Figure 4 in the revised manuscript) presenting the average SNR as a function of period. This figure clearly shows that the SNR of the correlation entering the inversion is actually good for all periods. The figure is also attached to this letter.

**Comment III) "*To improve the reliability of Rayleigh wave dispersions measurements we used the redundancy of the correlation tensor by using all components (RR, RZ, ZR, and ZZ) containing Rayleigh waves*". It is not clear that this really brings an improvement. In fact, you silently make the assumption that the Earth is isotropic. In case of azimuthal anisotropy, which surely**

**exists but may be difficult to constrain, you introduce a systematic error, e.g., by having Love waves on the vertical component and vice versa.**

Response III) The goal of the paper is to present an isotropic velocity model of the Eastern Alps. The combination of the various components containing Rayleigh waves improves the stability of the dispersions measurements by limiting the bias due to the non-uniform noise sources distribution. This type of combination is commonly done in Ambient Noise Tomography to improve the stability of the dispersions measurements.

**Comment IV) "*These parameters strongly affect the variance reduction of the final model. Stehly et al. (2009) recommended that the correlation length should be at least equal to grid size*". Since you explicitly say that these parameters affect the variance reduction, it is difficult to understand that you just fix them. How did you get to these values? For instance, did you run some kind of L-curve analysis for them?**

Response IV) we selected the inversion best parameters using L-curves analysis (lines 171-172). The variations of variance reduction as a function of damping factor and correlation length that are now provided as the new Figure 5 in the revised manuscript.

[revised manuscript text omitted]

---

## Referee Report (RR1)

The new version of the manuscript has largely dealt with the issues that I raised regarding the original manuscript. Although I have some comments, this study is in general well done and can be published after a minor revision.

Comments:
1) The tomographic results of both Rayleigh and Love wave at 30 and 40 s (Figure S5) are still not stable (white blanks). I think the authors may need to consider using larger damping and smoothing parameters at these periods.

2) Figure 5: There are two parameters used in the regularization, it will be better if you could indicate the fixed value of another parameter in the corresponding L-curve analysis.

3) Please check the reference, Barmin et al., 2001, 2012 are missing.

---

## Author Response (AR2)

Dear Editor,

We are pleased to submit the response to reviewer and editor's comments of "Crustal structures beneath the Eastern and Southern Alps from ambient noise tomography". We would like to thank again the editor and reviewer for their time spent on reviewing the manuscript. The comments were are valuable and very helpful for revising and improving our manuscript.

Points raised by the reviewer and editor have been carefully taken under consideration and addressed. In the following we list the reviewer and editor's comments in bold face followed by the authors' response to each of comments and the author's changes in manuscript. This letter is followed by a marked-up version of the revised manuscript in which all changes in the text have been highlighted.

Sincerely,
Ehsan Qorbani on behalf of the co-authors

**Referee's comments**

**Comment 1) You have not explained your choice of measuring group velocities instead of phase velocities. There isn't much advantage I can see to use group velocities in this case since you do not have to worry about the source function.**

Response 1) The choice was made for analysis of group velocities in this study, since our group has already gained experience with that established technique (e.g. the study of Schippkus et al., 2018 and 2020, to retrieve the structure of the Vienna basin region, as well as earlier studies). For the dataset at hand which includes temporary stations, the group velocities are also easier to measure, especially on the horizontal components where the correlations tend to be noisier, avoiding additional tests and processing steps needed to properly remove issues such as $2\pi$ jumps or wrong measurements which are more frequent with noisier data. For the dataset at hand we also believe that the choice of group or phase velocity is perhaps not the bottleneck. We agree with the Referee that the recent attempts to measure phase velocity are interesting. In future studies, we will probably use group and phase velocity simultaneously.

**Comment 2) The white blanks in Fig. S5 and in Fig. 8 are worrisome. Please, check what caused them and whether the inversions need to be rerun with different damping as suggested by the reviewer.**

Response 2) Even though we have carefully filtered the dispersion measurements in order to use only good quality measurements into the inversion, we have good ray coverage even at the longer periods (e.g. see Figure 3c and 3d). To map the inverted group velocities, we plot the cells that have at least three rays, which are those that have been used in the depth inversion. Therefore, the cells that do not meet this criteria are shown in white.
However, we thank the reviewer for the comment on this matter; it encouraged us to improve the number of good-quality measurements per cell. Different parameterizations (damping) have no influence on the appearance of the white blanks on the obtained group velocity maps, and we made an effort to increase the grid size in order to allow more rays to contribute to the inversion. Rerunning the inversion and testing different grid sizes, we

found that increasing the grid size by 33%, from 12 km to 16 km, could significantly improve the number of (good-quality) rays per cell without losing the resolution.

We therefore rerun both the group velocity and depth inversions. The final group velocity maps now have much less white blanks and also somewhat better coverage at the edges. We have updated figures 6 to 15, and Figure supplementary S3 and S5 accordingly. Please note that the new results represent the same structures as with the smaller 12 km grid size. This indicates that the velocity measurements are quite stable and sample the structures well, which is due to using only the good quality dispersion measurements. In addition, we would also like to mention that some of the white blanks showing up at the group velocity maps (Figure 8, 9) are because of plotting interpolation in order to provide readers a smooth view of the velocity contrasts. Figure 1 attached to this letter shows an example of group velocity map for Rayleigh wave at 20 sec with and without plotting interpolation.

**Comment 3) reference to Barmin et al., 2001, 2012 are missing**

Response 3) Barmin et al. (2001) has been cited in the manuscript (line 168, 198) and is listed in the references. The "Barmin 2012" mentioned in our earlier reply to the comments was a typo and mistakenly added. We apologize for the mistake.

[Figure]

Figure 1: Example of group velocity map at 20 sec with and without plotting interpolation. Part of the white blanks appearing in group velocity maps are due to plotting interpolation.

**Editor's comments**

**Comment 1) There should be a discussion of the effect of regularization on the inversion and how it may affect the anisotropy discussion. As you know, if the damping on the Love or the Rayleigh inversion were to be changed, the effect on the model may strongly affect the anisotropy results.**

Response 1) In this work, we inverted separately the Rayleigh and Love-wave group velocities which have well-resolved isotropic shear velocity models of the region. In addition, observing discrepancy between Rayleigh and Love shear velocity provides a hint to anisotropy.
We fully agree to the editor that inversion parameters can in general affect

the inverted velocity values, therefore, the Love/Rayleigh velocity ratio could also be affected. We have tested the effect of the different inversion parameters on the results. Figure 2 attached to this letter shows the variation of the Love/Rayleigh velocity ratio against different damping factors and correlation lengths. It shows that in our case inversion parameters have very little effect on the velocity ratio, which means the Rayleigh Love-wave discrepancy could be related to the presence of anisotropy. Moreover, we have observed the Rayleigh/Love-wave discrepancy not everywhere, but in certain areas where this indeed makes sense in terms of crustal tectonics, for instance to have shear deformation associated with lateral scape of the Alcapa block (see section 6.4 and Figure 1). However, to fully resolve the anisotropy, one should invert the Rayleigh and Love-wave dispersions simultaneously in order to obtain an anisotropic model of the region. We thank the editor for pointing out the effect of inversion regularization on Rayleigh and Love wave velocities and anisotropy. We have now added sentences mentioning that inversion parameters have no significant effect on the observed Rayleigh-Love wave discrepancies (Line 403-405). In addition to the section "Conclusion", we also mentioned in section 6.4 that joint inversion of the Rayleigh- and Love-wave dispersions is necessary to obtain anisotropic model of the region (Line 429-432).

[Figure]

Figure 2: Variation of Love/Rayleigh wave velocity ratio vs inversion parameters at different periods, a) damping factor and b) correlation length in km. This shows that inversion parameters have little effect on the velocity ratio.

**Comment 2) FTA is usually called FTAN in the literature. Please, change it everywhere in the text for consistency (with publications made over several decades).**

Response 2) We have replaced FTA with FTAN in the revised manuscript (line 139, 141).

[revised manuscript text omitted]